# Nuclear pore protein NUP210 depletion suppresses metastasis through heterochromatin-mediated disruption of tumor cell mechanical response

Ruhul Amin [1✉], Anjali Shukla[1], Jacqueline Jufen Zhu [2], Sohyoung Kim [3], Ping Wang[2], Simon Zhongyuan Tian[2], Andy D. Tran [1,4], Debasish Paul[1], Steven D. Cappell [1], Sandra Burkett [5], Huaitian Liu[1,6], Maxwell P. Lee [1,6], Michael J. Kruhlak[1,4], Jennifer E. Dwyer[7], R. Mark Simpson[7], Gordon L. Hager [3], Yijun Ruan[2] & Kent W. Hunter [1✉]

Mechanical signals from the extracellular microenvironment have been implicated in tumor and metastatic progression. Here, we identify nucleoporin *NUP210* as a metastasis susceptibility gene for human estrogen receptor positive (ER+) breast cancer and a cellular mechanosensor. *Nup210* depletion suppresses lung metastasis in mouse models of breast cancer. Mechanistically, NUP210 interacts with LINC complex protein SUN2 which connects the nucleus to the cytoskeleton. In addition, the NUP210/SUN2 complex interacts with chromatin via the short isoform of BRD4 and histone H3.1/H3.2 at the nuclear periphery. In *Nup210* knockout cells, mechanosensitive genes accumulate H3K27me3 heterochromatin modification, mediated by the polycomb repressive complex 2 and differentially reposition within the nucleus. Transcriptional repression in *Nup210* knockout cells results in defective mechanotransduction and focal adhesion necessary for their metastatic capacity. Our study provides an important role of nuclear pore protein in cellular mechanosensation and metastasis.

[1] Laboratory of Cancer Biology and Genetics, National Cancer Institute, NIH, Bethesda, MD, USA. [2] The Jackson Laboratory for Genomic Medicine, Farmington, CT, USA. [3] Laboratory of Receptor Biology and Gene Expression, National Cancer Institute, NIH, Bethesda, MD, USA. [4] Confocal Microscopy Core Facility, National Cancer Institute, NIH, Bethesda, MD, USA. [5] Molecular Cytogenetics Core Facility, National Cancer Institute, NIH, Frederick, MD, USA. [6] High-Dimension Data Analysis Group, National Cancer Institute, NIH, Bethesda, MD, USA. [7] Molecular Pathology Unit, Laboratory of Cancer Biology and Genetics, National Cancer Institute, NIH, Bethesda, MD, USA. ✉email: ruhul.amin@nih.gov; hunterk@mail.nih.gov

The majority of cancer-related mortality is due to distant metastasis, a process in which primary tumor cells invade into surrounding endothelium, evade immunosurveillance in the circulation, travel to a distant site, extravasate, and colonize a secondary site[1–3]. This highly inefficient process requires cancer cells to employ multiple genetic and epigenetic mechanisms to establish macroscopic lesions. However, due to the limited understanding of these mechanisms, it is difficult to target metastatic cells, even with the improved therapeutic strategies[4]. Hence, an improved understanding of the mechanisms that enable metastatic potential is crucial to improve patient outcomes.

Current studies have suggested that metastatic capacity in tumors may arise from the combined effect of acquired somatic mutations[5] and epigenetic changes influencing gene expression[6,7]. Changes in chromatin accessibility[8] or disruptions of large heterochromatin domains[9] can result in alteration of transcriptional programs that enable tumor cells to acquire metastatic phenotypes. In addition to these changes, previously our laboratory has demonstrated that inherited polymorphisms also have a significant effect on metastatic capacity[10]. Since the vast majority of inherited polymorphisms occur in noncoding DNA, these inherited variants are thought to contribute to phenotypes by altering gene expression rather than having direct consequences on protein function. A comprehensive understanding of how the genome mediates metastatic capacity will therefore require an understanding of not only the acquired events during tumor evolution but also how noncoding variants contribute to the complexities of gene regulation and transcriptional programs.

To gain further understanding of how polymorphisms affect regulatory elements and the subsequent metastatic phenotype, in this study, we have integrated chromatin accessibility and long-range chromatin interaction analysis to identify potential metastasis susceptibility genes with polymorphic promoters and/or distant enhancers. This analysis identified *Nup210*, a gene encoding a nuclear pore complex (NPC) protein, as a potential metastasis susceptibility gene. Although NPC proteins have recently been shown to be associated with several developmental disorders and cancers[11–14], their function in metastasis remains unexplored. Here, we established that NUP210 is responsive to mechanical signals of the extracellular microenvironment and promotes lung metastasis in mouse models of breast cancer through alteration of the mechanical response, focal adhesion, and cell migration in a nucleocytoplasmic transport-independent manner.

## Results

### Identification of *Nup210* as a candidate metastasis susceptibility gene.
To identify polymorphic accessible chromatin regions associated with metastatic colonization, benzonase-accessible chromatin (BACh) analysis was performed on isogenic cell lines derived from the spontaneous mammary tumor of BALB/cJ mice (67NR, 4T07, and 4T1)[15] (Fig. 1a). When orthotopically implanted into mice, 67NR forms tumors but remains localized, 4T07 cells disseminate to distant sites but rarely form macroscopic lesions, while 4T1 cells complete the metastatic process to form multiple pulmonary macroscopic lesions.

To identify transcriptional control regions enriched for the dormant-to-proliferative switch during metastatic colonization, 5303 BACh sites unique to the 4T1 cell line were identified. These sites were then intersected with the polymorphisms present in the eight founder strains[16] of the mouse Diversity Outbred (DO) genetic mapping panel[17]. Since the vast majority of these sites did not fall within the proximal promoters of protein-coding genes, chromatin interaction analysis through paired-end tag sequencing (ChIA-PET-seq) was performed in 4T1 cells to putatively link the polymorphic transcriptional control elements (promoters and/or enhancers). Finally, the resulting gene list was filtered through expression data of mice resulting from a cross between the MMTV-PyMT model[18] and the DO to identify genes associated with metastasis in this population. This strategy identified 52 potential metastasis susceptibility genes (Fig. 1b). To assess human relevance, the hazard ratios calculated for these genes in the mouse data were used to generate a weighted gene signature that was subsequently screened on human breast cancer gene expression datasets. Significant stratification of distant metastasis-free survival (DMFS) in estrogen receptor-positive (ER+) breast cancer was observed (Fig. 1c), consistent with a potential role of these genes in human breast cancer progression. NUP210, a nuclear pore protein, was selected for further validation due to the association of a number of previously identified metastasis susceptibility genes with the nuclear envelope[19]. To test the function of *Nup210* in metastasis, we initially asked whether variation in *Nup210* expression would correlate with metastatic potential. Our RNA-sequencing (RNA-seq) data from the 4T1 series of cell lines revealed that *Nup210* expression is higher in metastatic than nonmetastatic variant cell lines (Fig. 1d) suggesting a positive correlation of *Nup210* expression and metastatic potential.

### Polymorphisms in the *Nup210* promoter affect CTCF binding and *Nup210* transcription.
To examine the causal variant responsible for the transcriptional difference in the mouse genetic screen, the proximal promoter of *Nup210* was identified based on the BACh profile (Fig. 1e). Analysis of the Mouse Genomes Project database[16] revealed six single-nucleotide polymorphisms (SNPs) and one insertion/deletion (indel) in a 510 bp promoter region of *Nup210* (Fig. 1f). This included a 12 bp G-rich insertion in the *Nup210* promoter of the FVB/NJ, the MMTV-PyMT parental strain (Fig. 1g), located within a CCCTC-*binding* factor (CTCF)-binding region enriched in enhancer (H3K27Ac)-promoter (H3K4me3) marks[20]. CTCF binding on *Nup210* promoter also appeared to be evolutionarily conserved between mouse and human based on the analysis in MCF7 human breast cancer cell line (Fig. 1h). We initially tested whether the polymorphisms within *Nup210* promoters can affect gene expression via cloning these 510 bp regions into a promoter-luciferase reporter. A moderate (~10%), but significant, reduction of luciferase activity was observed for the FVB/NJ-derived *Nup210* promoter region compared to the BALB/cJ promoter indicating a functional impact of the polymorphism on *Nup210* transcription (Fig. 1i). Then, we examined the effect of 12 bp indel on CTCF binding and *Nup210* transcription as CTCF is known to regulate gene expression[21,22]. CTCF was preferentially enriched in the *Nup210* promoter region of 4T1 cells (derived from BALB/cJ mice) compared with 6DT1 cells (derived from FVB/NJ mice) (Fig. 1j). Consistently, the expression of *Nup210* and *Ctcf* were significantly higher in 4T1 cells than in 6DT1 cells (Fig. 1k). In addition, *Nup210* expression was significantly lower in the normal spleen of FVB/NJ compared to BALB/cJ mice (Fig. 1l) despite similar levels of Ctcf. CRISPR/Cas9-mediated deletion of CTCF-binding site on 4T1 *Nup210* polymorphic promoter showed significantly decreased *Nup210* mRNA, which further supported that *Nup210* promoter indel is a direct target of CTCF (Fig. 1m and Supplementary Fig. 1a–c). *Ctcf* mRNA was also downregulated in these deletion clones indicating a potential positive feedback loop between NUP210 and CTCF. Furthermore, *Ctcf* knockdown in 4T1 cells revealed a marked reduction of *Nup210* both at the transcript (Fig. 1n) and protein level (Fig. 1o), suggesting a direct effect of CTCF loss on *Nup210* transcription. Taken together, these results suggest that *Nup210* is a CTCF-regulated gene and

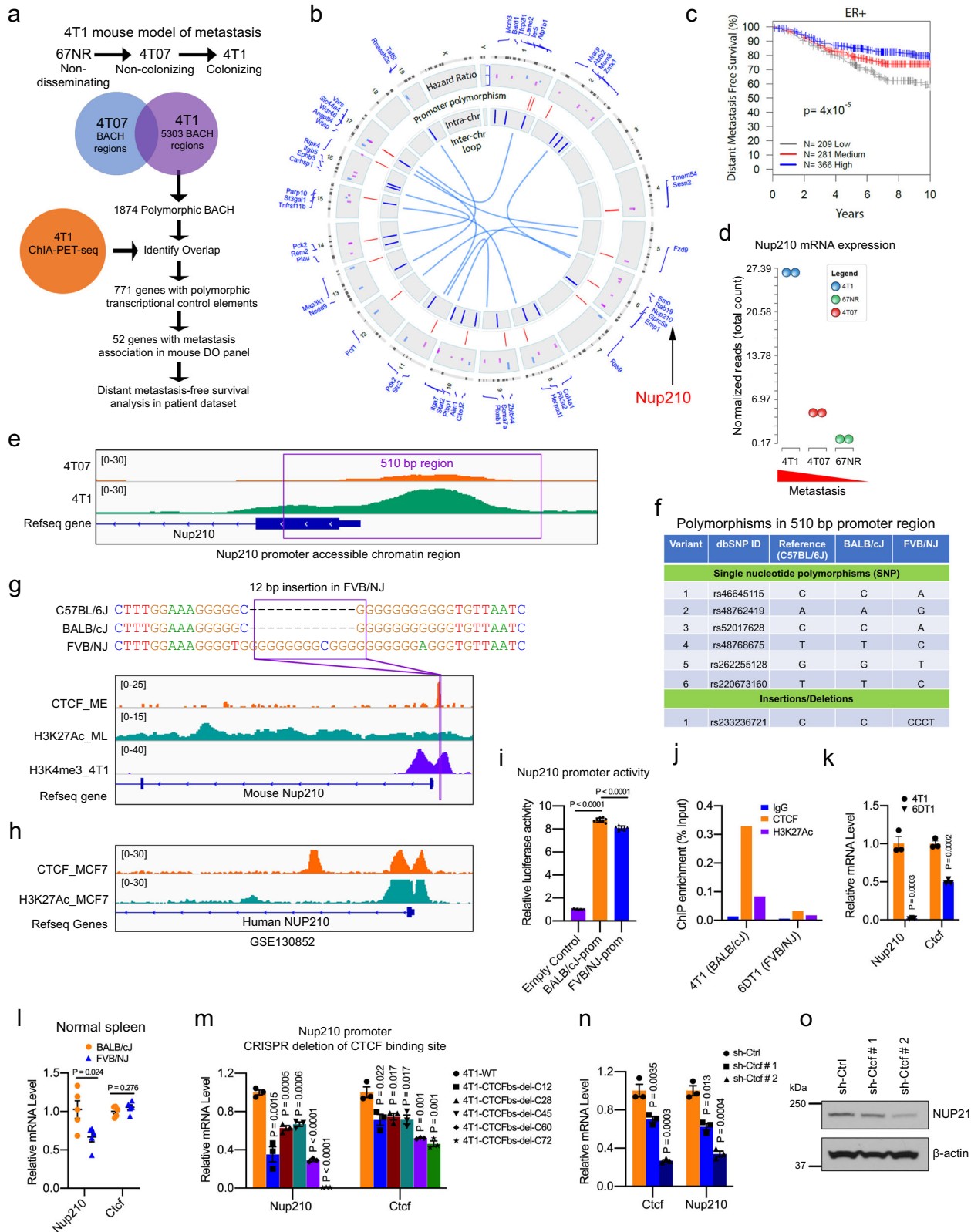

the indel in the promoter alters *Nup210* transcription by modulating CTCF binding to the promoter region.

**NUP210 expression is associated with metastasis in human breast cancer patients.** Consistent with the mouse data, *NUP210* amplification in human breast cancer patients was found to be associated with decreased overall survival in the Molecular

Taxonomy of Breast Cancer International Consortium (METABRIC) dataset[23] (Fig. 2a). Two independent human breast cancer gene expression datasets suggested that *NUP210* mRNA expression was significantly associated with reduced overall survival (Fig. 2b, METABRIC) and DMFS (Fig. 2c, Km plotter)[24] in estrogen receptor-positive (ER+) breast cancer patients. However, the patient survival outcome appears to be moderate in these datasets,

**Fig. 1 Genome-wide analysis of noncoding region polymorphisms identifies *Nup210* as a metastasis susceptibility gene. a** Scheme for metastasis susceptibility gene identification. **b** Circos plot representation of the result in (**a**). Red lines = polymorphic promoter, blue lines = intrachromosomal, and light blue lines = interchromosomal looping interactions. **c** Distant metastasis-free survival (DMFS) of human ER+ breast cancer patients stratified by their expression of the 52-gene signature. Kaplan–Meier analysis with the log-rank test. **d** *Nup210* mRNA expression in the mouse 4T1 cell line series. **e** Integrative Genomics Viewer (IGV) track of the *Nup210* promoter BACh region in 4T07 and 4T1 cells. **f** Polymorphisms within the 510 bp *Nup210* promoter region. **g** 12 bp FVB/NJ promoter indel is located within a CTCF-binding site in mouse *Nup210* promoter. **h** CTCF and H3K27Ac enrichment in human *NUP210* promoter of MCF7 cells. **i** Luciferase assay of BALB/cJ and FVB/NJ *Nup210* promoter regions. ANOVA, Tukey's multiple comparison test, mean ± s.e.m, n = 8 biological replicates. **j** ChIP analysis of CTCF and H3K27Ac at *Nup210* promoter of 4T1 and 6DT1 cells. **k** *Nup210* and *Ctcf* mRNA levels in cell lines (4T1, 6DT1). Two-tailed *t* test, mean ± s.e.m, n = 3 biological replicates. **l** Nup210 and Ctcf mRNA levels in mouse spleen (BALB/cJ or FVB/NJ). Two-tailed *t* test, mean ± s.e.m, n = 5 mice. **m** *Nup210* and *Ctcf* mRNA level in *Nup210* promoter CTCF-binding site-deleted clones. Two-tailed *t* test, mean ± s.e.m, n = 3 biological replicates. **n** Effect of *Ctcf* knockdown on 4T1 *Nup210* mRNA levels. Two-tailed *t* test, mean ± s.e.m, n = 3 biological replicates. **o** Effect of *Ctcf* knockdown on NUP210 protein level in 4T1 cells.

suggesting that *NUP210* expression is likely one of the multiple factors contributing to the patient outcome. Although the association of *NUP210* mRNA expression among triple-negative (ER−/PR−/HER2−) breast cancer patients were not consistently significant in these two datasets (Supplementary Fig. 2a, b), recent proteomics analysis revealed that higher NUP210 protein level is significantly associated with the poor DMFS of triple-negative cancer patient (Km plotter) (Supplementary Fig. 2c)[25]. METABRIC data suggest that *NUP210* expression is higher in luminal B and basal subtypes than in luminal A, Her2+, and claudin-low subtypes of breast cancer (Fig. 2d). Consistently, immunohistochemical analysis in the human normal mammary gland revealed that NUP210 is predominantly expressed in the luminal cell compartment (Fig. 2e). Analysis of a small human breast cancer tissue microarray (TMA) revealed that the NUP210 protein level was heterogeneous among the primary tumors of both ER+ and ER− patients (Fig. 2f). Interestingly, in addition to nuclear envelope staining of NUP210, cytoplasmic staining was also observed in many cases suggesting that NUP210 might be mislocalized in some patients. Confining the quantification of NUP210 signal intensity at the nuclear envelope revealed no significant difference between ER+ and ER− primary tumors (Fig. 2g). However, the NUP210 level was significantly higher in the lymph node metastases of ER+ than ER− patients (Fig. 2h, i), consistent with the survival outcome of the breast cancer patient in different datasets. Analysis in publicly available gene expression data of breast cancer patients revealed that visceral metastases (lung, liver) have significantly higher expression of *NUP210* than nonvisceral metastases (lymph node) (Fig. 2j). Furthermore, *NUP210* expression is significantly higher in metastases than primary tumor of prostate cancer (Supplementary Fig. 2d) and melanoma (Supplementary Fig. 2e). These data indicate that *NUP210* is associated with human cancer progression and further supports its potential role as a metastasis susceptibility gene.

**Depletion of *Nup210* in cancer cells decreases lung metastasis in mice.** To validate the role of *Nup210* in metastasis, luminal-like orthotopic mammary tumor transplantation models were used[26]. Three different cell lines were utilized for this analysis: 4T1[15], 6DT1, derived from the mammary tumor of FVB/MMTV-MYC transgenic mouse, and MVT1, derived from the mammary tumor of FVB/MMTV-MYC/VEGF double transgenic mouse[27]. Primary tumors derived from orthotopically implanted *Nup210* short hairpin RNA knockdown (Fig. 3a and Supplementary Fig. 3a) cells showed variability in tumor weight, with decreased tumor weight for 4T1 cells (Fig. 3b), but increased tumor weights for both 6DT1 and MVT1 cells (Supplementary Fig. 3b). However, *Nup210* knockdown resulted in a decrease in pulmonary metastases for all three cell lines (Fig. 3c, d and Supplementary Fig. 3c–f), and this decrease was preserved after normalizing metastasis counts by tumor weight to account for the variability in primary tumor growth. These results were therefore consistent

with the association of *NUP210* with poor prognosis in patients with ER+ breast cancers.

To further validate the role of *Nup210* in metastatic progression, CRISPR/Cas9-mediated knockout (KO) of *Nup210* was performed in the 4T1 cell line (Fig. 3e). Similar to short hairpin RNA (shRNA) result, *Nup210* KO in two different clones, N9 and N13, diminished primary tumor weight and lung metastases (Fig. 3f–h). Although we observed differential effects of *Nup210* loss on primary tumor weight, cell cycle analysis on *Nup210* KO 4T1 cells (Fig. 3i, j) revealed no consistent changes in the distribution of cell population among cell cycle stages, suggesting that the effect of NUP210 loss on tumor and metastasis is independent of a cell-intrinsic ability of NUP210 to regulate proliferation. Finally, 4T1 cells with overexpression of *Nup210* showed no difference in primary tumor weight (Fig. 3k, l), but lung metastasis was significantly increased (Fig. 3m, n). Taken together, these results indicate that *Nup210* promotes lung metastasis in mouse models of luminal breast cancer.

**NUP210 tethers histone H3.1/3.2 to the nuclear periphery.** To investigate the mechanism by which NUP210 alters metastatic function, initially we examined the effect of NUP210 depletion on nucleocytoplasmic transport, a general function of the nuclear pore. A recent study has suggested that loss of NUP210 does not affect the general nucleocytoplasmic transport in differentiating myotubes[28]. Consistent with this, transfection of NES-tdTomato-NLS nucleocytoplasmic transport reporter (Supplementary Fig. 4a) into *Nup210* knockdown 4T1 cells with or without the nuclear export inhibitor leptomycin B showed no significant differences in nucleocytoplasmic transport of the reporter protein (Supplementary Fig. 4b).

To better understand how NUP210 promotes metastasis, NUP210 coimmunoprecipitation (Co-IP)-mass spectrometry analysis was performed (Fig. 4a). Both endogenous and myc-tagged NUP210 potentially interact with multiple chromatin-associated molecules including histone H3.1 (Fig. 4b). We decided to examine NUP210–histone H3.1 interaction due to the potential association of H3.1 with poor outcome in breast cancer patients (P = 0.0642) in METABRIC dataset (Fig. 4c) and reported mutations of histone H3.1 in human cancer[29,30]. Reciprocal Co-IP in 4T1 cells using an antibody that recognizes both H3.1 and H3.2 validated the predicted H3.1–NUP210 interaction (Fig. 4d, e). H3.1/3.2 also pulled down Lamin B1, a component of the nuclear lamina, suggesting that this interaction occurs at the nuclear periphery. Reciprocal co-IP using both endogenous and Flag-tagged H3.1 in human 293FT cells also showed a similar result (Fig. 4f). Immunofluorescence analysis confirmed the association of H3.1/3.2 and NUP210 at the nuclear periphery (Fig. 4g). Extending the analysis in human breast cancer cells revealed that NUP210–H3.1/3.2 interaction is more prominent in ER+ MCF7 cells than ER− MDA-MB-231 cells

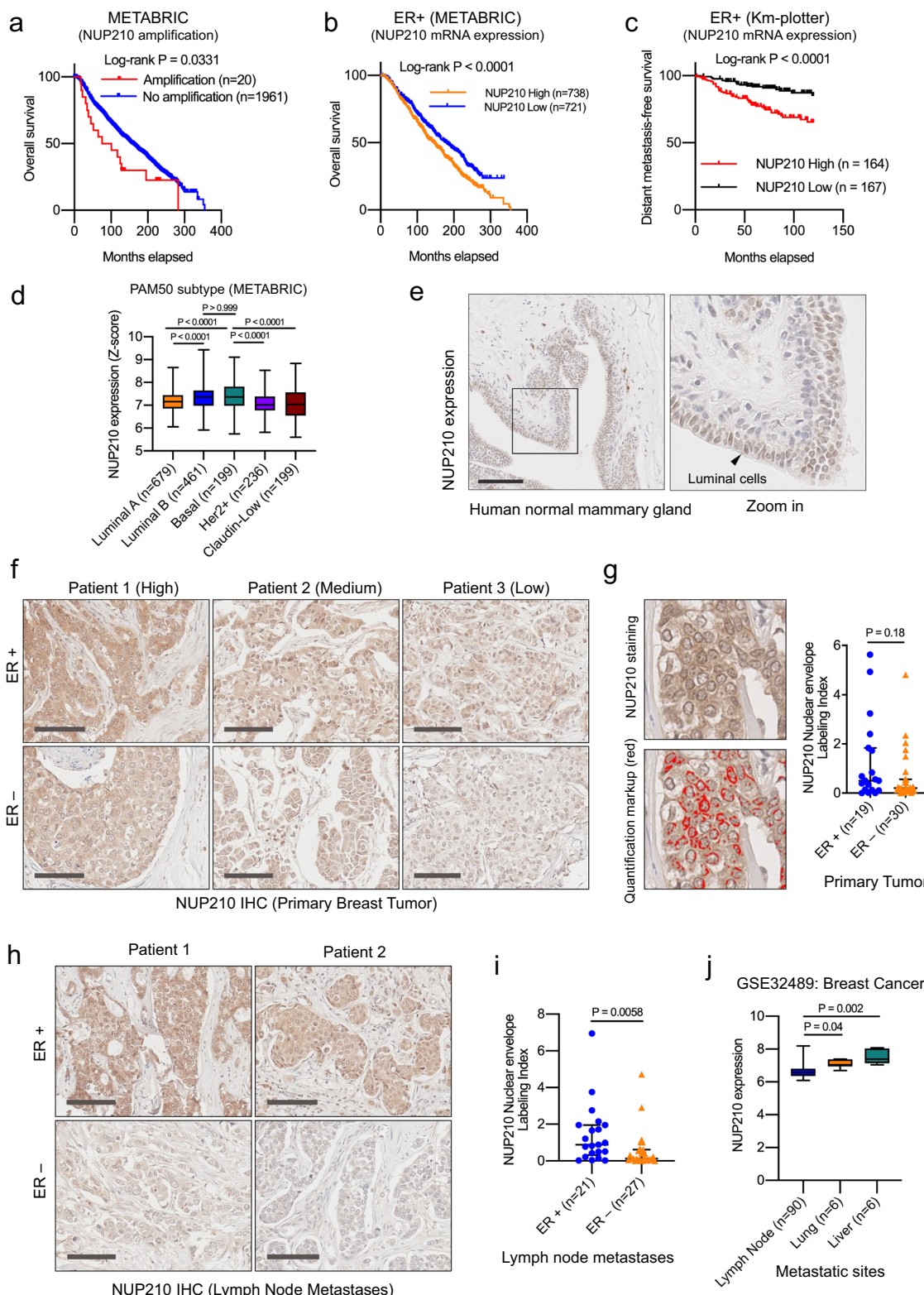

(Supplementary Fig. 5a, b). These results establish an evolutionarily conserved association of NUP210 with H3.1/3.2 in mouse and human cells.

**Increased recruitment of heterochromatin-modifying enzymes, SUV39H1 and EZH2, to H3.1/3.2 in *Nup210* KO cells.** H3.1 is known to be incorporated into nucleosomes in the S phase of the cell

cycle in a replication-dependent manner[31], which suggested that NUP210 loss might affect S-phase progression. As mentioned earlier, there were no consistent changes in the S phase of the cell cycle in *Nup210* KO cells (Fig. 3i, j), suggesting that NUP210–H3.1/3.2 association might have a different role in metastatic cells. The effect of NUP210 loss on H3.1/3.2 interactions in metastatic cells was therefore examined. Loss of NUP210 resulted in a redistribution of H3.1/3.2 from the nuclear periphery to foci-like structures inside the

**Fig. 2 *NUP210* expression is associated with poor outcome in human ER+ breast cancer patients. a** Association of *NUP210* amplification on the overall survival (OS). **b** Association of *NUP210* mRNA level on the OS and **c** DMFS in ER+ patients. **d** *NUP210* mRNA level within METABRIC dataset. Kruskal–Wallis ANOVA with Dunn's multiple comparison test. The box extends from 25th to 75th percentile, whiskers extend from smallest values to the largest values, and the horizontal line represents median. '*n*' in *X*-axis is the number of patients in each subtype. **e** NUP210 protein expression in the human normal mammary gland. Scale bar = 100 μm. **f** Immunohistochemistry and nuclear envelope quantification **g** of NUP210 in the primary tumors of ER+ and ER− cancer patients. Scale bar = 100 μm. Mann–Whitney *U* test, error bar represents median with interquartile range. **h** Immunohistochemistry analysis and nuclear envelope quantification. **i** NUP210 protein in the lymph node metastases of ER+ and ER− patients. Scale bar = 100 μm. Mann–Whitney *U* test, error bar represents median with interquartile range. **j** *NUP210* mRNA level in multiple human breast cancer metastatic sites. Kruskal–Wallis ANOVA with Dunn's multiple comparison test. The box extends from 25th to 75th percentile, whiskers extend from smallest values to the largest values, and the horizontal line represents median. '*n*' in *X*-axis is the number of patients analyzed per condition.

nucleus (Fig. 4h, i), suggesting that NUP210 might be tethering H3.1/3.2 to the nuclear envelope. A significant enrichment of the H3K27me3 heterochromatin mark at the nuclear periphery was also observed. Costaining with H3K9me3, a pericentric heterochromatin-associated histone marker, and 4′,6-diamidino-2-phenylindole (DAPI) revealed that the redistributed H3.1/3.2 colocalized with heterochromatin in *Nup210* KO cells (Fig. 4j, k). 3D reconstruction revealed that H3.1/3.2 foci were localized on the periphery of H3K9me3-marked region (Fig. 4l and Supplementary Movies 1 and 2). Many of the foci appeared on the heterochromatin-enriched nucleolar periphery (Fig. 4m). Like 4T1, H3.1/3.2 was also redistributed from the nuclear periphery to intranuclear foci in *NUP210*-depleted ER+ MCF7 cells (Supplementary Fig. 5c, d). Although there was a significant loss of H3.1/3.2 from the nuclear periphery of *NUP210*-depleted, ER−, MDA-MB-231 cells (Supplementary Fig. 5e, f), H3.1/3.2 foci was not prominent in these cells suggesting a subtype-specific effect of NUP210 loss. Live-cell imaging demonstrated that nuclear size was significantly decreased in *Nup210* KO cells throughout the cell cycle (Fig. 4n), suggesting that the loss of NUP210 might increase heterochromatinization[32]. Co-IP in *Nup210* KO cells revealed a decreased association of H3.1/3.2 with Lamin B1 and increased association with heterochromatin-modifying enzymes, SUV39H1 (catalyzing H3K9me3) (Fig. 4o) and EZH2 (Fig. 4p), a key component of the polycomb-repressive complex 2 (PRC2) (catalyzing H3K27me3)[33]. Further analysis of other PRC2 components revealed an increased association of EZH2 with SUZ12 (Fig. 4q) and H3K27me3 (Fig. 4r) in *Nup210* KO cells. Taken together, these results suggest that NUP210 restricts the recruitment of heterochromatin-modifying enzymes to H3.1/3.2 at the nuclear periphery.

**NUP210 loss increases heterochromatin spreading and decreases expression of cell adhesion/migration-related genes.** To test whether NUP210 loss promotes heterochromatin formation, we performed chromatin immunoprecipitation followed by sequencing (ChIP-seq) analysis of NUP210, H3K27me3 (repressive), and H3K4me3 (active) histone marks. ChIP-seq analysis revealed that the majority (55.33%) of the NUP210 peaks are within intergenic regions (Supplementary Fig. 6a), with 19.01% of NUP210 peaks within 10 kb of transcription start sites (TSS). The rest of the peaks were found within gene bodies, but mainly enriched in 3′-transcription end sites of the genes (Supplementary Fig. 6b). Moreover, 40% (9962 peaks) of NUP210 peaks overlap with H3K27me3 (false discovery rate (FDR) < 0.05) (Supplementary Fig. 6c), indicating H3K27me3 enrichment surrounding the NUP210-bound gene bodies. In addition, in *Nup210* KO cells, increased H3K27me3 over NUP210-bound gene bodies was observed (Fig. 5a), suggesting that these genes were undergoing heterochromatinization. There was an overall expansion of H3K27me3 peak regions, suggesting H3K27me3-marked heterochromatin spread (Fig. 5b). Integration of ChIP-seq dataset for active enhancer (H3K27Ac)[34] revealed that NUP210 was enriched at regions surrounding H3K27Ac (Supplementary Fig. 6d).

Loss of NUP210 resulted in an increase in the repressive H3K27me3 mark across these enhancer regions, suggesting NUP210 might be preventing heterochromatin from spreading across distal regulator elements as well as gene bodies.

Since enhancer–promoter interactions are critical for the regulation of gene expression[35,36] and H3.1/3.2 is preferentially enriched at the poised promoters, we applied structured illumination microscopy (SIM) to examine the intranuclear distribution of H3K4me3 (promoter) and H3K27Ac (enhancer) marks in *Nup210* KO cells. Although there were no significant changes in H3K4me3 volume in *Nup210* KO cells, H3K27Ac volume was significantly reduced (Supplementary Fig. 6e, f and Supplementary Movies 3 and 4), suggesting the decrease of enhancer regions. There was no noticeable alteration of H3K4me3 distribution in relation to H3.1/3.2 foci in *Nup210* KO cells (Supplementary Fig. 6g). However, ChIP-seq of H3K4me3 revealed that the TSS of 1199 genes showed 2-fold decreased enrichment of H3K4me3 in *Nup210* KO cells (Fig. 5c). RNA-seq of *Nup210* KD 4T1 cells identified 282 downregulated (2-fold) and 249 upregulated (2-fold) genes (Supplementary Fig. 6i), 62 of which had H3K4me3 loss at the promoter and ≥2-fold decreased expression (Fig. 5d) and thus were likely to be direct targets of NUP210 regulation. Gene ontology (GO) analysis on the 62 genes showed enrichment of cell migration and chemotaxis processes (Fig. 5e), including chemokines (*Ccl2*, *Cxcl1*, and *Cxcl3*), cytokine (*Il-1a*), and cell adhesion (*Postn*, *Serpine1*, *Itga7*, *Itgb2*) molecules (Fig. 5f and Supplementary Fig. 6j). Similar transcriptional changes were also observed in NUP210-depleted 6DT1 cell line (Supplementary Fig. 6k). Integration of publicly available high-throughput chromosome conformation capture (Hi-C) data from mouse embryonic stem cells (mESCs)[37] using the 3D-Genome Browser[38] revealed that H3K27me3 accumulated across the *Cxcl* and *Ccl2* loci topologically associated domains (TADs) in *Nup210* KO cells (Fig. 5g), and was accompanied by a marked loss of H3K4me3 at the promoters of these genes, implying that NUP210 loss results in heterochromatinization across the *Cxcl*/*Ccl2* enhancer and promoter regions.

To test whether the expression of NUP210-regulated genes can be rescued in *Nup210* KO cells by inhibiting heterochromatization, cells were treated with an EZH2 inhibitor GSK126, which prevents H3K27me3 modification. Treatment with GSK126 significantly increased the expression of *Ccl2*, *Cxcl1*, *Cxcl3*, *Postn*, and *Il-1a* (Fig. 5h), suggesting that they were suppressed through heterochromatinization in *Nup210* KO cells. In contrast, GSK126 treatment decreased the expression of *Itga7*, *Itgb2*, and *Serpine1*, implying that they were either not directly suppressed by heterochromatinization or could be suppressed by a nonspecific effect of the drug. Interestingly, perinuclear distribution of H3.1/3.2 and nuclear size was significantly restored in GSK126-treated *Nup210* KO cells (Fig. 5i, j). Live-cell tracking of nuclear size further supported the observation that GSK126 treatment significantly rescued the nuclear size of *Nup210* KO cells (Fig. 5k). These results are consistent with the role of NUP210 in preventing H3K27me3 heterochromatinization of perinuclear DNA.

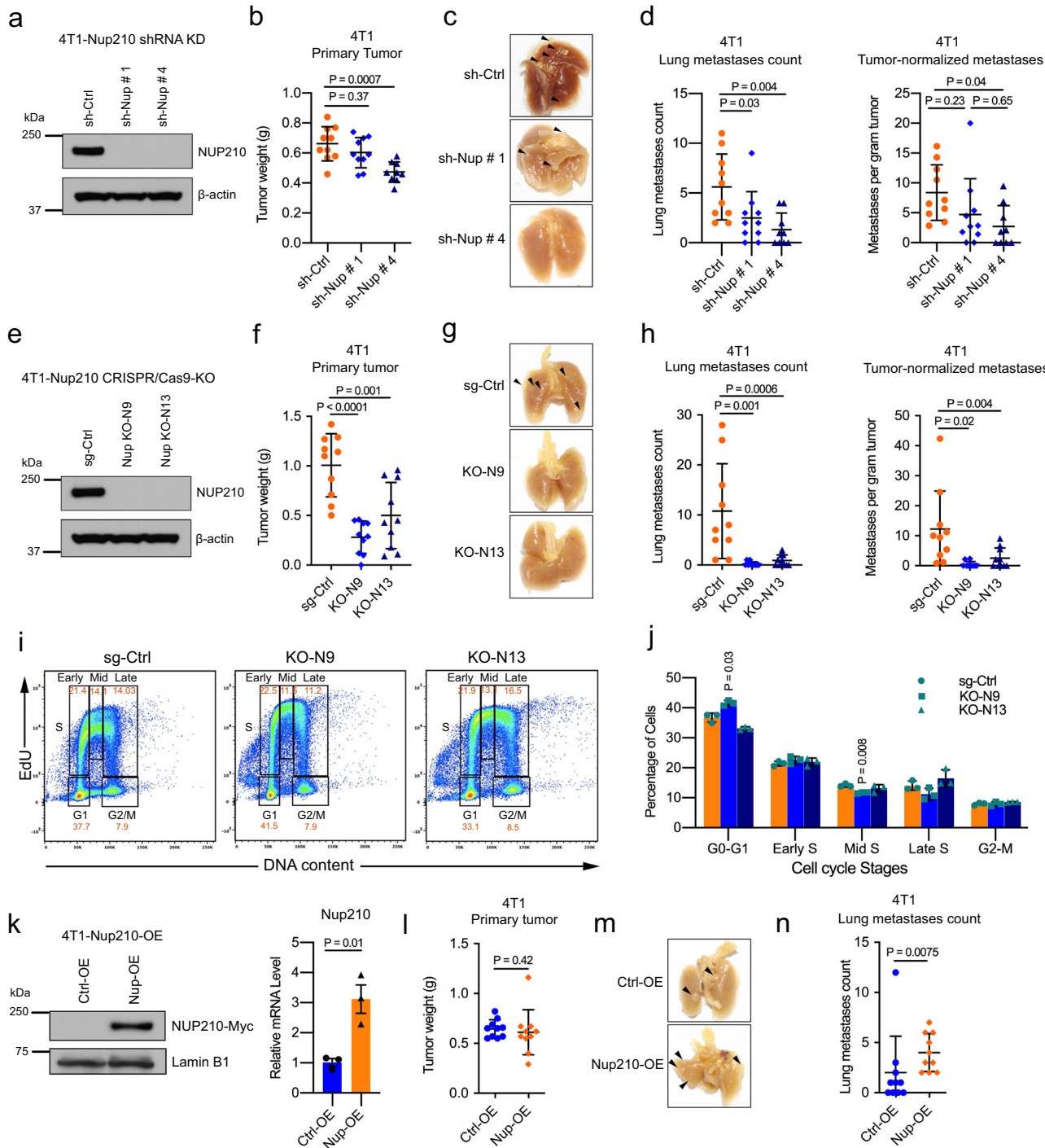

**Fig. 3 Depletion of *Nup210* in 4T1 metastatic cancer cells reduces lung metastasis in mice. a** Western blot of *Nup210* knockdown (KD) in 4T1 cells. **b** Primary tumor weight, **c** representative lung images, and **d** lung metastases count and lung metastases count normalized to primary tumor weight in orthotopically transplanted *Nup210* KD 4T1 cells. ANOVA with Tukey's multiple comparison test, mean ± s.d. sh-Ctrl, $n = 10$; sh # 1, $n = 10$; sh # 4, $n = 9$ mice. **e** Western blot showing CRISPR/Cas9-mediated knockout (KO) of *Nup210* in 4T1 cells. **f** Primary tumor weight, **g** representative lung images, **h** lung metastases count and tumor-normalized metastases count after orthotopic transplantation of *Nup210* KO 4T1 cells. ANOVA with Tukey's multiple comparison test, mean ± s.d. $n = 10$ mice per group. **i** DNA content analysis with EdU incorporation of the different stages of the cell cycle in sg-Ctrl and *Nup210* KO 4T1 cells. **j** Quantification of cell cycle stage distribution in sg-Ctrl and *Nup210* KO 4T1 cells. Two-tailed *t* test, mean ± s.e.m, $n = 3$ biological replicates. **k** Western blot (left) and qRT-PCR (right) of NUP210 overexpression in 4T1 cells. Two-tailed *t* test, mean ± s.e.m, $n = 3$ biological replicates. **l** Primary tumor weight, **m** representative lung images, and **n** lung metastases count after orthotopic transplantation of NUP210-overexpressing 4T1 cells. Mann–Whitney *U* test, mean ± s.d. $n = 10$ mice per group.

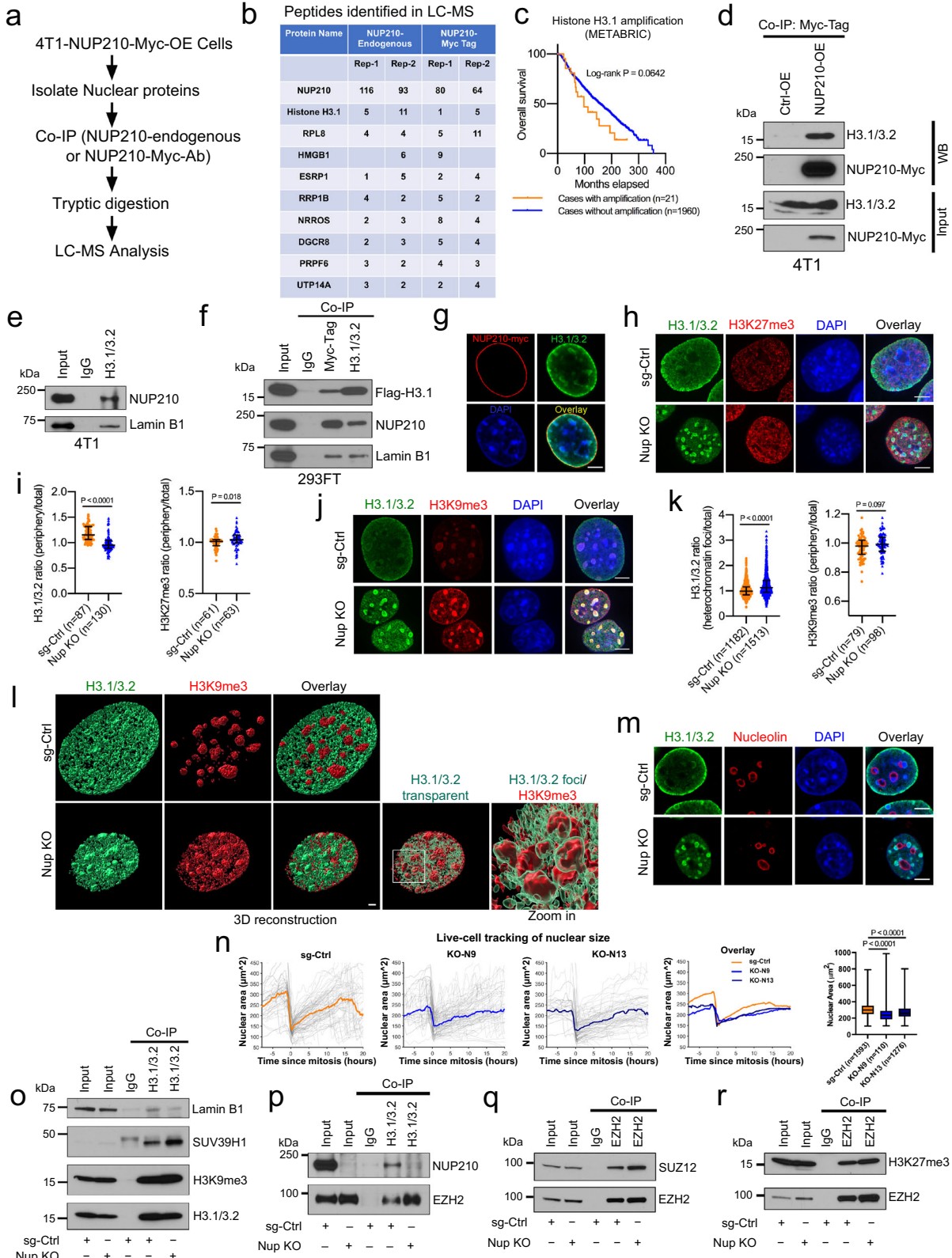

**Nup210 loss is associated with differential repositioning of cell migration-related gene loci.** To determine whether NUP210 regulates its target genes through anchoring TADs to the nuclear periphery, 3D-DNA fluorescent in situ hybridization (FISH) was performed. FISH probes were generated from bacterial artificial chromosome clones (~2 Mb) spanning *Cxcl*, *Postn*, *Ccl2*, and *Itgb2* genomic regions. As expected, the majority of the FISH spots were observed near the nuclear periphery in aneuploid 4T1 cells (Supplementary Fig. 7a). Interestingly, in *Nup210* KO cells, *Cxcl*, *Postn*, and *Itgb2* loci were repositioned within the nucleus (Supplementary Fig. 7b), with *Cxcl* and *Itgb2* loci repositioned closer to the nuclear periphery, and the *Postn* locus repositioned away from the periphery. Radial positioning of the *Ccl2* locus, however, was not significantly changed. Rather, the

**Fig. 4 H3.1/3.2 interacts with NUP210 and is redistributed to heterochromatin foci upon NUP210 loss. a** NUP210 coimmunoprecipitation (Co-IP)-LC-MS analysis. **b** Peptide count in LC-MS analysis with endogenous and Myc-tag-NUP210 antibodies. **c** Association of histone H3.1 amplification on the overall survival of breast cancer patients. **d** Co-IP of NUP210-Myc with H3.1/3.2 in 4T1 cells. **e** Co-IP of endogenous H3.1/3.2, NUP210, and Lamin B1 in 4T1 cells. **f** Reciprocal Co-IP of Flag-H3.1 and NUP210-Myc in HEK293FT cells. **g** Representative immunofluorescence images of H3.1/3.2 and NUP210-Myc localization in 4T1 cells. Scale bar = 5 μm. **h** Representative images of H3.1/3.2 and H3K27me3 in sg-Ctrl and *Nup210* KO 4T1 cells. Scale bar = 5 μm. **i** H3.1/3.2 and H3K27me3 nuclear periphery vs total nuclear intensity ratio in *Nup210* KO 4T1 cells. Mann–Whitney *U* test, error bar represents median with interquartile range. '*n*' equals the number of cells analyzed. **j** Representative images of H3.1/3.2 localization with H3K9me3 in *Nup210* KO 4T1 cells. Scale bar = 5 μm. **k** (Left) H3.1/3.2 (heterochromatin foci vs total nucleus) and (Right) H3K9me3 (periphery vs total nuclear intensity) intensity ratios in *Nup210* KO 4T1 cells. Mann–Whitney *U* test, error bar represents median with interquartile range. '*n*' equals the number of cells analyzed. **l** 3D reconstruction of H3.1/3.2 and H3K9me3 distribution in *Nup210* KO 4T1 cells. The intensity was adjusted to visualize distinct foci. Scale bar = 2 μm. **m** Representative images of H3.1/3.2 distribution at the nucleolar (nucleolin) periphery in *Nup210* KO 4T1 cells. Scale bar = 5 μm. **n** Live-cell imaging of nuclear size of *Nup210* KO 4T1 cells before and after mitosis. Thin line, single nuclear size traces; thick line, median of nuclear size, box plot represents average nuclear size. '*n*' equals the number of cells analyzed per condition. Kruskal–Wallis ANOVA with Dunn's multiple comparison test. The box represents 25th to 75th percentile, the whiskers represent the data range, and the horizontal line represents the median. **o** Co-IP of H3.1/3.2 with Lamin B1, SUV39H1, and H3K9me3 in *Nup210* KO 4T1 cells. **p** Co-IP of H3.1/3.2 with EZH2; **q** EZH2 and SUZ12; **r** EZH2 and H3K27me3 in *Nup210* KO 4T1 cells.

distance of *Ccl2* FISH spots to heterochromatin foci was significantly reduced in *Nup210* KO (Supplementary Fig. 7c) cells. These results suggest that NUP210 loss is associated with altered chromatin topology and nuclear positioning of cell migration-related gene loci. Since three-dimensional chromatin domains are organized into transcriptionally active (A compartment) and repressive (B compartment) compartments within the nucleus[39], repositioning of these genes in *Nup210* KO cells is consistent with their repositioning into repressive B compartment due to the heterochromatinization.

**NUP210 is mechanosensitive and regulates focal adhesion, cell migration, and invasion.** Alterations in cell adhesion/migration-related gene expression can lead to changes in cellular adhesion, a critical phenotype in metastatic progression. *Nup210* KO cells exhibited less spread, round morphology when they were grown on type I collagen and fibronectin matrices (Fig. 6a). Staining for phospho-FAK (Y397) and phalloidin revealed that loss of *Nup210* significantly decreased the focal adhesions (area and count), actin stress fibers, and cell spreading when plated on type I collagen or fibronectin (Fig. 6b, c). Similar results were obtained in the case of *Nup210*-knockdown 4T1 (Supplementary Fig. 8a), 6DT1 (Supplementary Fig. 8b, c), and human MDA-MB-231 cells (Supplementary Fig. 8d). Consistently, the protein level of phospho-FAK (Y397) was also decreased in NUP210-depleted mouse (Supplementary Fig. 8e–g) and human (Supplementary Fig. 8h, i) breast cancer cells. In addition, there was a significant decrease of phospho-FAK (Y397) staining within the *Nup210* KO primary tumors, suggesting that NUP210 loss also affects focal adhesion in vivo (Fig. 6d). To test whether the constitutively active FAK can rescue the metastasis defects of *Nup210* KO cells, we stably expressed Myc-tagged FAK in *Nup210* KO cells (Fig. 6e). Although there was no significant effect on primary tumors in FAK-expressing cells (Fig. 6f), both tumor-normalized lung metastases count (Fig. 6h) and metastasis incidence (Fig. 6i) were significantly rescued indicating that NUP210 is mediating its effects on metastasis through the focal adhesion signaling pathway.

As focal adhesion contributes to the mechanical response of tumor cells, we asked whether NUP210 is mechanosensitive in metastatic cells. 4T1 cells grown on soft/low stiffness (0.2 kPa) condition exhibited a round colony-like morphology, while on the higher stiffness conditions (12 kPa), these cells exhibited a more spread morphology (Fig. 7a). Increased protein levels of p-FAK (Y397) and NUP210 were found in the higher stiffness condition (Fig. 7b), indicating that NUP210 protein is mechanosensitive in 4T1 cells. Moreover, NUP210 protein level was higher on cells grown on type I collagen than fibronectin (Supplementary Fig. 8j), suggesting that NUP210 might be

sensitive to different ECM compositions. Many of the NUP210-regulated genes were also sensitive to higher ECM stiffness conditions (Fig. 7c). However, unlike protein level, transcript levels of Nup210, FAK (Ptk2), as well as a known mechanosensitive transcription factor, Yap1, were not significantly upregulated in higher stiffness conditions. This result is consistent with the previous observation that soft matrices tend to degrade mechanosensitive proteins[40] and rapid reduction of NUP210 protein on soft matrix could be due to the post-transcriptional loss of protein stability. NUP210-depleted cells exhibited decreased spreading on the higher stiffness condition (Fig. 7d), consistent with NUP210 contributing to mechanosensation. As mechanosensation can occur through alteration of actin dynamics[41,42], treatment with an actin polymerization inhibitor, cytochalasin D (CytoD), in 4T1 cells demonstrated that NUP210-depleted cells had decreased recovery of actin polymerization, as marked by phalloidin staining of actin stress fibers (Supplementary Fig. 8k). As the mechanical response is mediated through alteration of actomyosin tension, staining with phospho-myosin light chain 2 (MLC2-S19) and phalloidin revealed a decreased association of F-actin and activated myosin II (phospho-MLC2-S19) in *Nup210* KO cells grown on type I collagen and fibronectin matrices (Supplementary Fig. 8l). To further investigate the actomyosin contractility of *Nup210*-depleted cells, we stably expressed a fluorescently tagged stochiometric F-actin (F-tractin) and myosin II activity reporter (MLC2)[43] in 6DT1 cells and performed live-cell imaging analysis. On type I collagen, both F-tractin and MLC2 remained associated at the actin stress fibers and lamellipodia in control cells compared to *Nup210* knockdown cells where they were localized at the nuclear periphery (Fig. 7e). In CytoD-treated conditions, F-tractin and MLC2 association was disrupted. After CytoD washout, control cells quickly recovered and maintained their F-tractin–MLC2 association at the lamellipodia (Supplementary Movie 5). In contrast, this association was disrupted in *Nup210* knockdown cells and MLC2 mainly localized at the nuclear periphery (Supplementary Movie 6). These results suggest that the loss of NUP210 is associated with decreased actomyosin contractility of cancer cells.

As actomyosin contractility is critical for cell migration and invasion, live-cell imaging analysis of cell migration was performed and revealed a significant decrease of cell migration in *Nup210* KO cells than sg-Ctrl cells (Fig. 7f and Supplementary Movies 7–9). A significant reduction of cell invasion was also observed in NUP210-depleted 4T1 (Fig. 7g) and 6DT1 cells (Fig. 7h). Taken together, these results demonstrate that NUP210 is part of a sensor of the extracellular matrix (ECM) stiffness and composition that significantly affects the migratory and invasive ability of tumor cells.

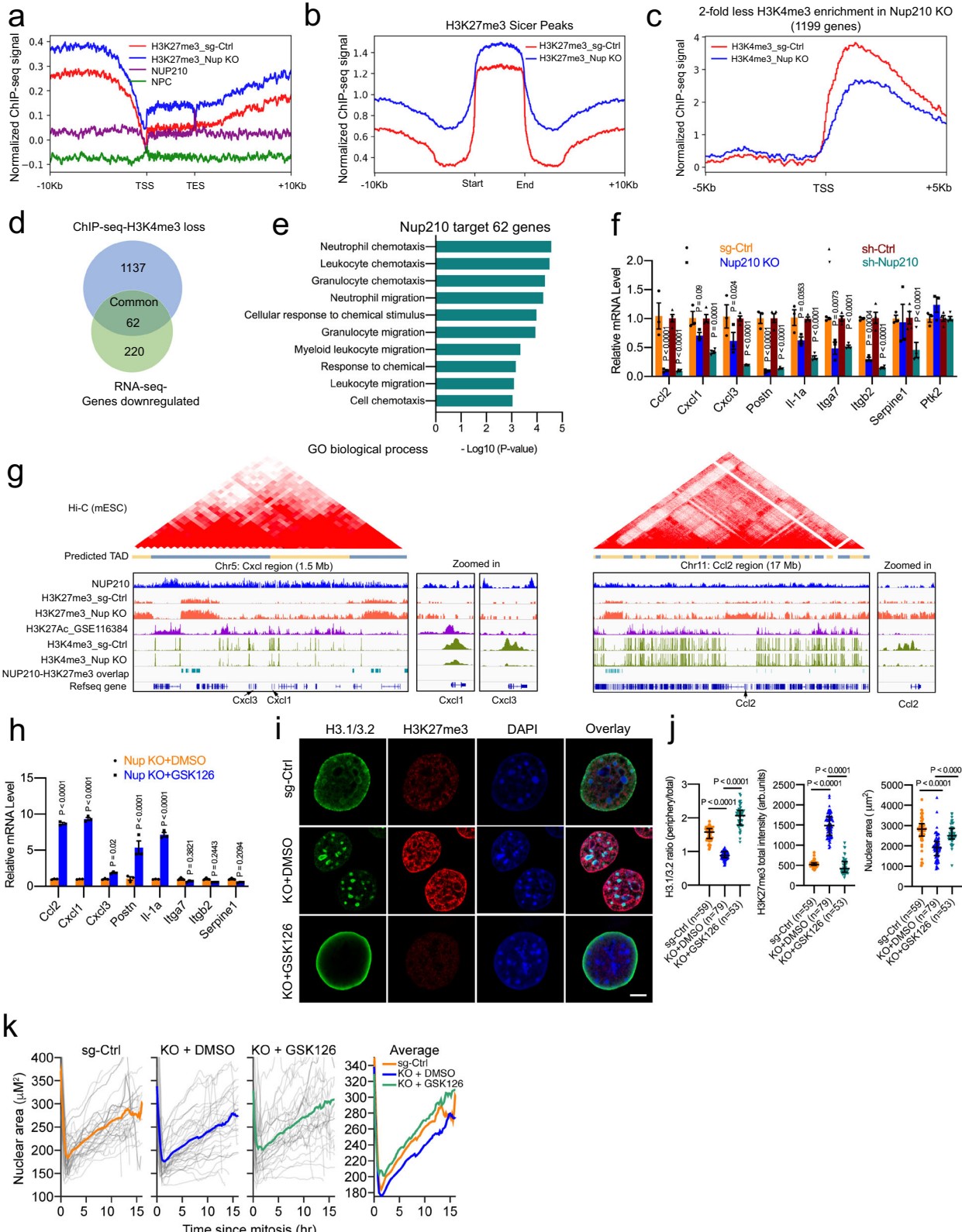

**NUP210 is a LINC (linker of nucleoskeleton and cytoskeleton) complex-associated protein and regulates mechanosensitive gene expression.** As mechanotransduction is mediated through the nuclear envelope-anchored LINC complex proteins[42,44], we asked whether NUP210-dependent mechanical response is mediating through its association with LINC complex proteins. Co-IP revealed that NUP210 preferentially interacts with LINC

complex protein SUN2, but not SUN1 (Fig. 8a). Immuno-fluorescence imaging showed that NUP210 was associated with SUN2 at the nuclear periphery (Fig. 8b). Interestingly, the perinuclear localization of SUN1 and SUN2 was significantly disrupted in *Nup210* KO cells (Fig. 8c). SUN2 was distributed diffusely throughout the cells with a significant increase in intranuclear distribution in *Nup210* KO cells (Fig. 8d). In

**Fig. 5 Loss of NUP210 is associated with H3K27me3-marked heterochromatin spreading in 4T1 cells. a** H3K27me3 peak enrichment on gene bodies in *Nup210* KO cells within NUP210-enriched regions. **b** H3K27me3 ChIP-seq profile in sg-Ctrl and *Nup210* KO cells. **c** H3K4me3 ChIP-seq profile on gene promoters in sg-Ctrl and *Nup210* KO cells. **d** Overlap of genes with H3K4me3 loss on the promoter and downregulated expression in *Nup210* KD cells. **e** Gene Ontology (GO) analysis of overlapped genes from (**d**). **f** qRT-PCR analysis of cell migration-related genes in *Nup210* KO and KD 4T1 cells. Multiple two-tailed *t* test, mean ± s.e.m, *n* = 3 biological replicates (sg-ctrl vs Nup210 KO), *n* = 4 biological replicates (sh-Ctrl vs sh-Nup210). **g** Representative ChIP-seq (NUP210-, H3K27Ac-, H3K27me3- and H3K4me3) tracks with predicted TADs within *Cxcl* and *Ccl2* region. **h** qRT-PCR analysis of cell migration-related genes in DMSO- and GSK126-treated *Nup210* KO 4T1 cells. Multiple two-tailed *t* test, mean ± s.e.m, *n* = 3 biological replicates. **i** Representative images of H3.1/3.2 and H3K27me3 distribution in DMSO- and GSK126-treated *Nup210* KO cells. Scale bar = 5 μm. **j** H3.1/3.2 intensity ratio, H3K27me3 intensity and nuclear area quantification in DMSO- and GSK126-treated *Nup210* KO 4T1 cells. Kruskal–Wallis ANOVA with Dunn's multiple comparison test, error bar represents median with interquartile range. 'n' in *X*-axis is the number of cells analyzed per condition. **k** Live-cell tracking of a nuclear area in DMSO- and GSK126-treated *Nup210* KO cells. *n* = 40 (sg-Ctrl), *n* = 41 (KO + DMSO), and *n* = 31 cells analyzed per conditions.

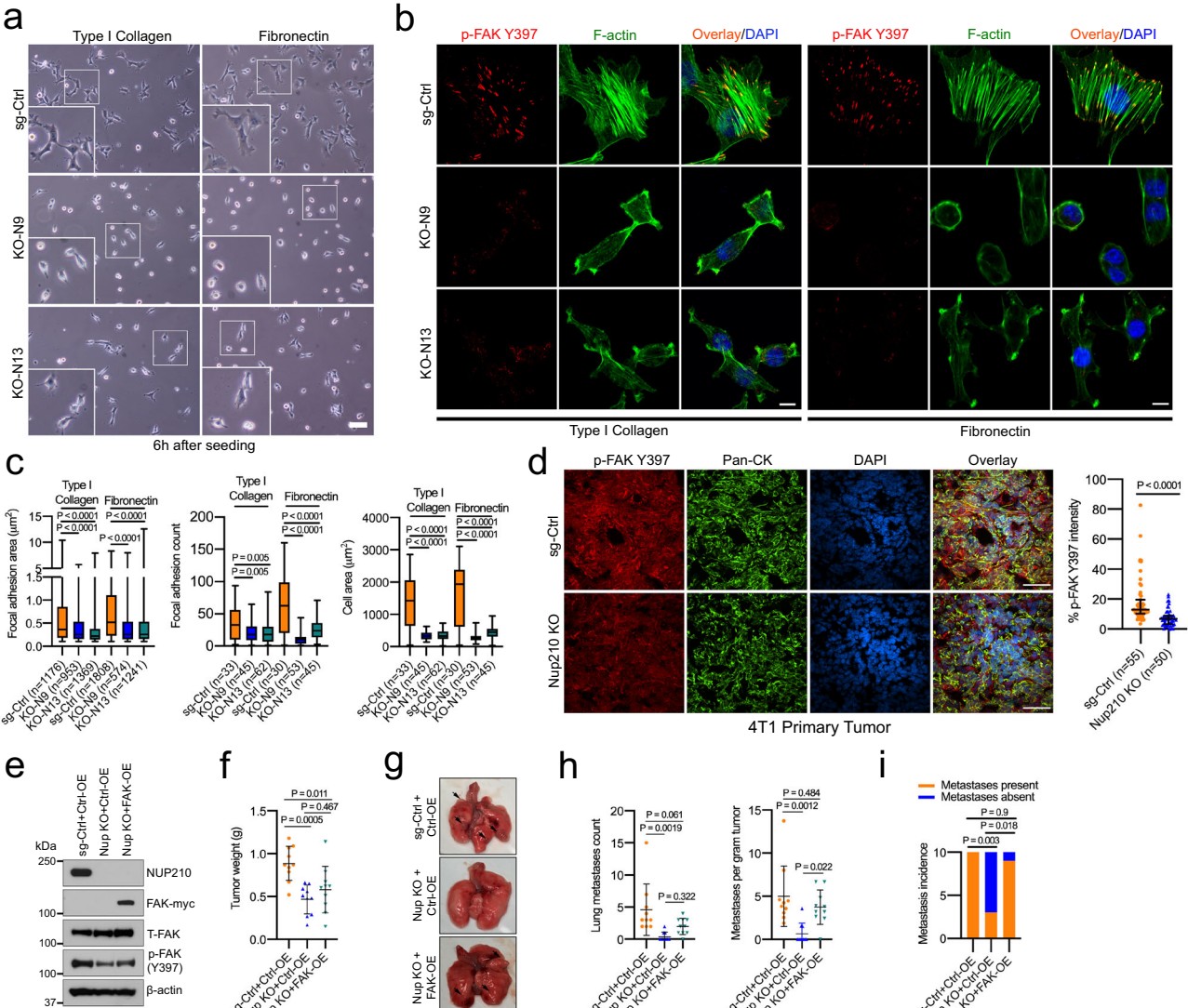

**Fig. 6 *Nup210* regulates focal adhesion of tumor cells. a** Representative brightfield images of 4T1 *Nup210* KO cells on type I collagen and fibronectin. Scale bar = 100 μm. **b** Representative images of p-FAK (Y397) focal adhesions and F-actin in sg-Ctrl and *Nup210* KO cells on type I collagen and fibronectin. Scale bar = 10 μm. **c** Quantification of focal adhesion area, count, and cell spreading in sg-Ctrl and *Nup210* KO 4T1 cells. ANOVA with Tukey's multiple comparison correction. The box represents the 25th to 75th percentile, whiskers represent data range, and the horizontal line represents the median. **d** (Left) Immunostaining of *Nup210* KO (KO-N13) 4T1 primary tumors with p-FAK Y397 and pan-cytokeratin (Pan-CK) antibodies. Scale bar = 50 μm. (Right) quantification of p-FAK Y397 signal in the Pan-CK-stained tumor area. *N* = 5 mice per condition, ~10 independent fields per mouse tumor section. Mann–Whitney *U* test, median with interquartile range. **e** Western blot of myc-tagged FAK overexpression in *Nup210* KO 4T1 cells. **f** Primary tumor weight in FAK-overexpressing *Nup210* KO (KO-N13) 4T1 cells. ANOVA with Tukey's multiple comparison correction, mean ± s.d. *n* = 10 mice per group. **g** Representative lung images of the mice injected with FAK-overexpressing *Nup210* KO cells. **h** Lung metastasis count and metastases normalized to tumor weight in FAK-overexpressing *Nup210* KO cells. ANOVA with Tukey's multiple comparison correction, mean ± s.d. *n* = 10 mice per group. **i** Metastasis incidence in mice injected with FAK-overexpressing *Nup210* KO cells. $\chi^2$ test with Bonferroni correction. *n* = 10 mice per group.

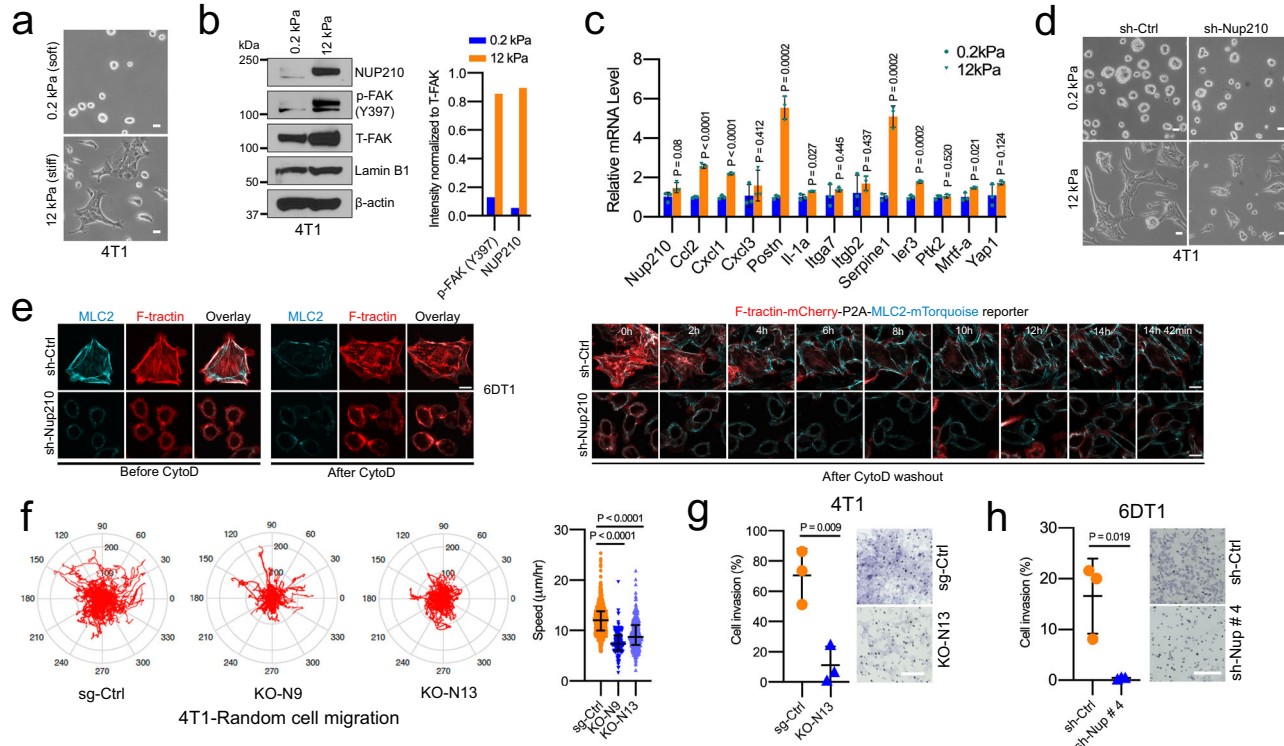

**Fig. 7 NUP210 is mechanosensitive and regulates tumor cell migration and invasion. a** Representative images of 4T1 cells on fibronectin-coated hydrogel layers of soft (0.2 kPa) and stiff (12 kPa) matrices. Scale bar = 10 μm. **b** (Left) Western blot of NUP210, p-FAK Y397, and T-FAK (Total FAK) proteins of 4T1 cells grown on soft or stiff matrices. (Right) Quantification of signals normalized to T-FAK intensity. **c** qRT-PCR of *Nup210*-regulated and known mechanosensitive genes (*Ier3*, *Mrtf-a*, and *Yap1*) genes in soft and stiff matrices. Multiple two-tailed *t* test, mean ± s.e.m. *n* = 3 biological replicates. **d** Morphology of sh-Ctrl and *Nup210* KD 4T1 cells in soft and stiff matrices. Scale bar = 10 μm. **e** Live-cell imaging of F-tractin-MLC2 reporter transduced 6DT1 *sh-Nup210* cells. Cells were treated with cytochalasin D for 1 h and imaged overnight after drug washout. Scale bar = 10 μm. **f** (Left) cell migration tracks (red) of sg-Ctrl and *Nup210* KO 4T1 cells. (Right) Quantification of cell speed. Kruskal–Wallis ANOVA with Dunn's multiple comparison correction, sg-Ctrl (*n* = 847 cells), KO-N9 (*n* = 124 cells), KO-N13 (*n* = 375 cells), error bar represents median with interquartile range. **g** Quantification (left) and representative images (right) of cell invasion for 4T1 (at 48 h) and (**h**) 6DT1 (at 24 h) NUP210-depleted cells. Two-tailed *t* test, mean ± s.d. *n* = 3 biological replicates. Scale bar = 100 μm.

addition, the ratio of the nuclear periphery to total nuclear intensity was significantly decreased in the case of both SUN2 and SUN1. Since NUP210 was affecting the LINC complex, we looked at the nuclear lamina of *Nup210* KO cells more closely. Although there were no significant changes in total Lamin B1 intensity, total Lamin A/C intensity was significantly increased in *Nup210* KO cells (Fig. 8e, f). In contrast, the ratio of the nuclear periphery to the total intensity of both Lamin B1 and Lamin A/C was significantly reduced in *Nup210* KO cells. More interestingly, 3D surface reconstruction revealed that both Lamin B1 and Lamin A/C appeared as crevices inside the nucleus of *Nup210* KO cells (Fig. 8g and Supplementary Movies 10 and 11). Since increased Lamin A/C level has previously been shown to increase nuclear stiffness resulting in decreased cell migration and tumorigenesis[45,46], we measured the level of total Lamin A/C protein using western blot. However, the total Lamin A/C protein level was not significantly changed in *Nup210* KO cells (Supplementary Fig. 9a), suggesting that the increased Lamin A/C intensity in immunofluorescence was probably due to the heterochromatin-mediated decrease of nuclear size, which resulted in a higher concentration of Lamin A/C at the nuclear envelope. Subcellular fractionation revealed that SUN2 and Lamin B1 proteins are mislocalized in the cytoplasmic fraction of *Nup210* KO cells (Fig. 8h). There was an overall decrease of SUN1 protein in the nuclear fraction of *Nup210* KO cells. The nuclear translocation of known mechanosensitive proteins (MRTF-A and YAP) was not affected in *Nup210* KO cells (Fig. 8h). Rather, there

was an overall decrease of MRTF-A/MKL1 in nuclear fraction and the level of YAP was decreased in both cytoplasmic and nuclear fraction of *Nup210* KO cells. This result suggests that the role of NUP210 in mechanosensation is independent of the canonical nuclear translocation mechanism of mechanosensitive proteins[41,47]. However, the overall distribution of nuclear pore was not affected by *Nup210* KO (Supplementary Fig. 9b). In addition, the dynamic organization of nuclear pore was not compromised in NUP210-depleted cells as confirmed by fluorescence recovery after photobleaching (FRAP) analysis of GFP-tagged nuclear pore protein POM121 (Supplementary Fig. 9c). These results suggested that the loss of NUP210 is associated with alteration of nuclear lamina and uncoupling of LINC complex machinery resulting in altered mechanical response, cell migration, and metastasis.

We then investigated the link between the alteration of mechanotransduction machinery and transcriptional suppression of *Nup210* KO cells. We have previously shown that metastasis susceptibility genes RRP1B and the short isoform of BRD4 (BRD4-SF) interact with LINC complex proteins at the nuclear lamina[19,48]. Similarly, BRD4 has been linked to mechanotransduction[49] and metastasis[50]. So, we hypothesized that NUP210 is probably associated with this protein complex and mediates the interaction with H3.1/3.2-associated chromatin to regulate the gene expression in metastasis. Co-IP revealed that endogenous histone H3.1/3.2 was associated with NUP210, BRD4-SF, SUN2, and RRP1B protein complex in 4T1 cells

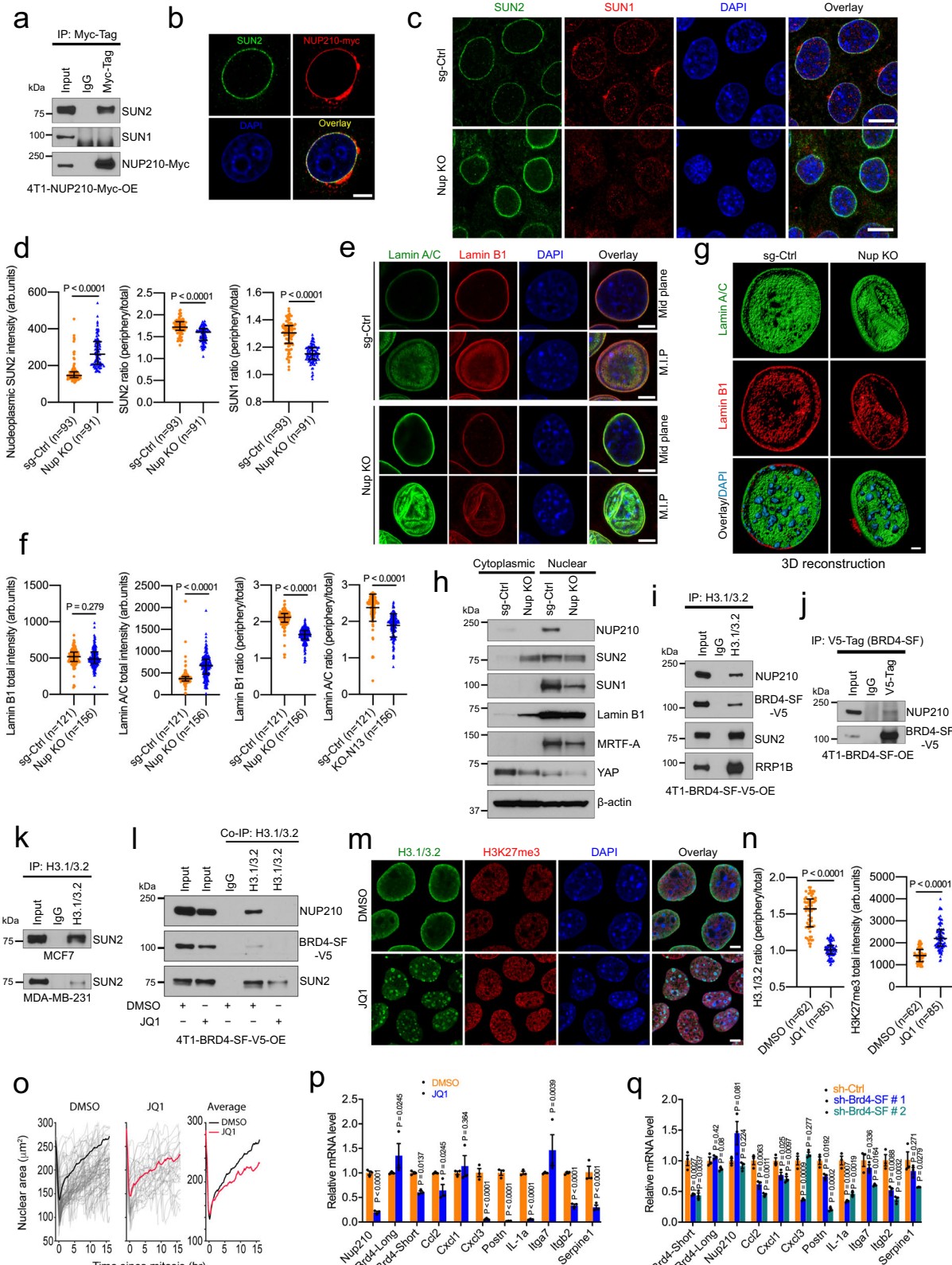

(Fig. 8i). BRD4-SF also interacts with NUP210 in these cells (Fig. 8j). H3.1/3.2–SUN2 interaction was also evident in human breast cancer cells MCF7 and MDA-MB-231 (Fig. 8k). We then tested the necessity of either NUP210 or BRD4-SF in bridging the interaction between the LINC complex machinery and H3.1/3.2 on chromatin. Although the interaction of SUN2 with H3.1/3.2 was unperturbed in *Nup210* KO cells (Supplementary

Fig. 10a), complete disruption of H3.1/3.2–NUP210 interaction and partial disruption of H3.1/3.2–SUN2 interactions were observed in bromodomain inhibitor JQ1-treated 4T1 cells (Fig. 8l). This result suggested that BRD4-SF is crucial for bridging the interactions among the molecules in this putative complex. Interestingly, similar to *Nup210* KO, redistribution of H3.1/3.2 from the nuclear periphery to the interior and increased

**Fig. 8 NUP210 is a LINC complex-associated protein mediating chromatin connection through the short isoform of BRD4. a** Coimmunoprecipitation of myc-tagged NUP210 with SUN1 and SUN2. **b** Colocalization of SUN2 and myc-tagged NUP210. Scale bar = 5 μm. **c** Distribution of SUN1 and SUN2 in *Nup210* KO (KO-N13) 4T1 cells. Scale bar = 10 μm. **d** Quantification of SUN2 and SUN1 intensity in *Nup210* KO 4T1 cells. Mann–Whitney *U* test, error bar represents median with an interquartile range. '*n*' equals the number of cells analyzed. **e** Distribution of Lamin B1 and Lamin A/C in *Nup210* KO 4T1 cells. MIP maximum intensity projection. Scale bar = 5 μm. **f** Quantification of Lamin B1 and Lamin A/C intensity in *Nup210* KO 4T1 cells. Mann–Whitney *U* test, error bar represents median with an interquartile range. '*n*' equals the number of cells analyzed. **g** 3D reconstruction of Lamin B1, Lamin A/C, and DAPI (heterochromatin foci) distribution in *Nup210* KO 4T1 cells. Scale bar = 2 μm. **h** Subcellular fractionation of LINC complex proteins and mechanosensitive MRTF-A and YAP in *Nup210* KO 4T1 cells. **i** Co-IP of H3.1/3.2 with LINC complex proteins SUN2, BRD4 short isoform (BRD4-SF), and RRP1B in 4T1 cells. **j** Co-IP of V5-tagged BRD4-SF with NUP210 in 4T1 cells. **k** Co-IP of H3.1/3.2 and SUN2 in human MCF7 and MDA-MB-231 cell line. **l** Co-IP of H3.1/3.2 with NUP210 and SUN2 in JQ1-treated 4T1 cells. **m** Distribution of H3.1/3.2 and H3K27me3 in 4T1 cells treated with bromodomain inhibitor JQ1. Scale bar = 5 μm. **n** Quantification of H3.1/3.2 and H3K27me3 intensity in JQ1-treated 4T1 cells. Mann–Whitney *U* test, error bar represents median with an interquartile range. '*n*' equals the number of cells analyzed. **o** Live-cell tracking of nuclear size in DMSO control (*n* = 122) and JQ1-treated 4T1 cells (*n* = 127). **p** qRT-PCR of NUP210-regulated genes in JQ1-treated 4T1 cells. Two-tailed *t* test, mean ± s.e.m. *n* = 3 biological replicates. **q** qRT-PCR of NUP210-regulated genes in BRD4-SF knockdown 4T1 cells. Two-tailed *t* test, mean ± s.e.m. *n* = 3.

heterochromatin mark H3K27me3 was observed in bromodomain inhibitor JQ1-treated cells (Fig. 8m, n). Live-cell imaging of nuclear size also revealed that JQ1 treatment significantly decreased nuclear size (Fig. 8o). Decreased expression of *Nup210* and some of the *Nup210*-dependent genes was found in JQ1-treated cells (Fig. 8p). Similar gene expression changes were also observed in BRD4-SF-specific knockdown 4T1 cells (Fig. 8q). Taken together, NUP210 is a LINC complex-associated protein and suggests that NUP210 is a regulator of mechanosensitive gene expression program through interacting with BRD4-SF-H3.1/3.2-associated chromatin.

To determine the cause and effect relationship of NUP210-dependent response, we treated 4T1 cells with actin polymerization inhibitor CytoD and a global heterochromatin promoting H3K27me3-specific histone demethylase inhibitor (GSKJ4). Actomyosin organization was drastically reduced in CytoD-treated cells as shown by F-actin and p-MLC2-S19 immunostaining (Supplementary Fig. 10b). Unlike CytoD, GSKJ4 had a minor effect on F-actin stress fibers, but it disrupted p-MLC2-S19 distribution, suggesting the alteration of actomyosin tension in GSKJ4-treated cells. Treatment with both of these drugs resulted in the redistribution of histone H3.1/3.2 from the nuclear periphery to heterochromatin foci, similar to the effect seen in *Nup210* KO cells (Supplementary Fig. 10c, d). Furthermore, CytoD treatment showed additional effects such as micronuclei and double nuclei formation. Treatment of cells with GSKJ4 resulted in a similar transcriptional profile as Nup210-depleted cells, which was not recapitulated by the treatment with CytoD (Supplementary Fig. 10e). *Nup210* mRNA level was slightly decreased in CytoD-treated cells, but unchanged at the protein level (Supplementary Fig. 10f). Despite the unchanged level of *Nup210* mRNA in GSKJ4-treated cells, a significant decrease of NUP210 protein level was observed, suggesting post-transcriptional downregulation of *Nup210* (Supplementary Fig. 10f). Moreover, decreased levels of focal adhesion (p-FAK, T-FAK) and LINC complex protein (SUN2) were found in the GSKJ4-treated condition. Decreased Lamin B1 protein levels by both drugs indicated that they are also capable of disrupting the nuclear lamina. Taken together, these results suggest a positive feedback loop between NUP210-dependent heterochromatin regulation and actin cytoskeletal tension, and the change in focal adhesion is likely to be the consequence of disrupting this feedback loop (Supplementary Fig. 10g).

**NUP210 regulates circulating tumor cells (CTCs) in mice.**
Many of the NUP210-regulated genes are secretory molecules like cytokines (IL-1α), chemokines (CCL2, CXCL1, CXCL3), and adhesion molecules (POSTN, SERPINE1), some of which have known prometastatic roles in breast cancer[51–54]. Cytokine array

analysis revealed decreased secretion of cytokines (IL-1α), chemokines (CCL2, CXCL1), and adhesion molecules (POSTN, SERPINE1) in *Nup210* knockdown 4T1 cells (Fig. 9a). To investigate whether NUP210 mediates its effect on focal adhesion and cell migration through transcriptional control of secretory molecules, *Ccl2* was chosen as a candidate and examined for NUP210 KO-associated phenotypes. Knockdown of *Ccl2* in 4T1 cells (Fig. 9b) resulted in decreased p-FAK (Y397) (Fig. 9c), decreased focal adhesion, and cell spreading (Fig. 9d, e). Moreover, a significant decrease in cell migration was observed upon *Ccl2* knockdown phenocopying the loss of NUP210 in these cells (Fig. 9f).

We then tested whether recombinant CCL2 can rescue the spreading defect of *Nup210* KO cells. Treating with recombinant CCL2 for 5 h moderately rescued the spreading phenotype of *Nup210* KO cells (Fig. 9g). To test whether CCL2 can rescue the metastatic defect of *Nup210* KO cells, we overexpressed *Ccl2* in *Nup210* KO cells through lentiviral delivery. However, only ~25% of *Ccl2* expression was rescued (Fig. 9h). Injecting these *Ccl2*-overexpressing cells into mice did not have a significant effect on primary tumors (Fig. 9i). Although *Ccl2* expression in *Nup210* KO cells showed an increased trend of lung metastasis, it was not statistically significant (Fig. 9j, k). This result suggests that either the level of the rescue of *Ccl2* expression in *Nup210* KO cell was not sufficient to have a significant effect on metastasis or rescuing with a single gene from the NUP210 transcriptional network cannot fully recover the NUP210 loss.

As *Nup210*-depleted cancer cells exhibited adhesion defects and decreased the level of secretory molecules necessary for tumor cell adhesion to endothelial wall[55], we speculated that NUP210 loss might affect the CTC levels. Therefore, based on CD45−/CK+ staining, putative CTCs were isolated 28 days after orthotopic implantation of 6DT1 cells[56]. Under these conditions, approximately ~0.5–1% of the cells in the blood of 6DT1-sh-Ctrl tumor-bearing mice were CD45−/CK+ putative CTCs. In contrast, CTC counts in mice with *Nup210*-depleted tumors were indistinguishable from the FVB tumor-free control (Fig. 9l). These results suggest that the effect on focal adhesion and cell migration in NUP210-depleted cells is mediated through the NUP210-dependent transcriptional control of cytokine/chemokine secretory pathways that are required for tumor cell extravasation or for the survival of cells within the bloodstream.

In summary, our model suggests that growth factors and integrins transmit the mechanical signals via NUP210–SUN1–SUN2–BRD4-SF–H3.1/3.2 protein complex to regulate mechanosensitive genes (Fig. 9m), which in turn activate downstream signaling pathways necessary for focal adhesion, cell migration, and metastasis. In *Nup210* KO cells, this signaling cascade is disrupted, thereby decreasing metastasis.

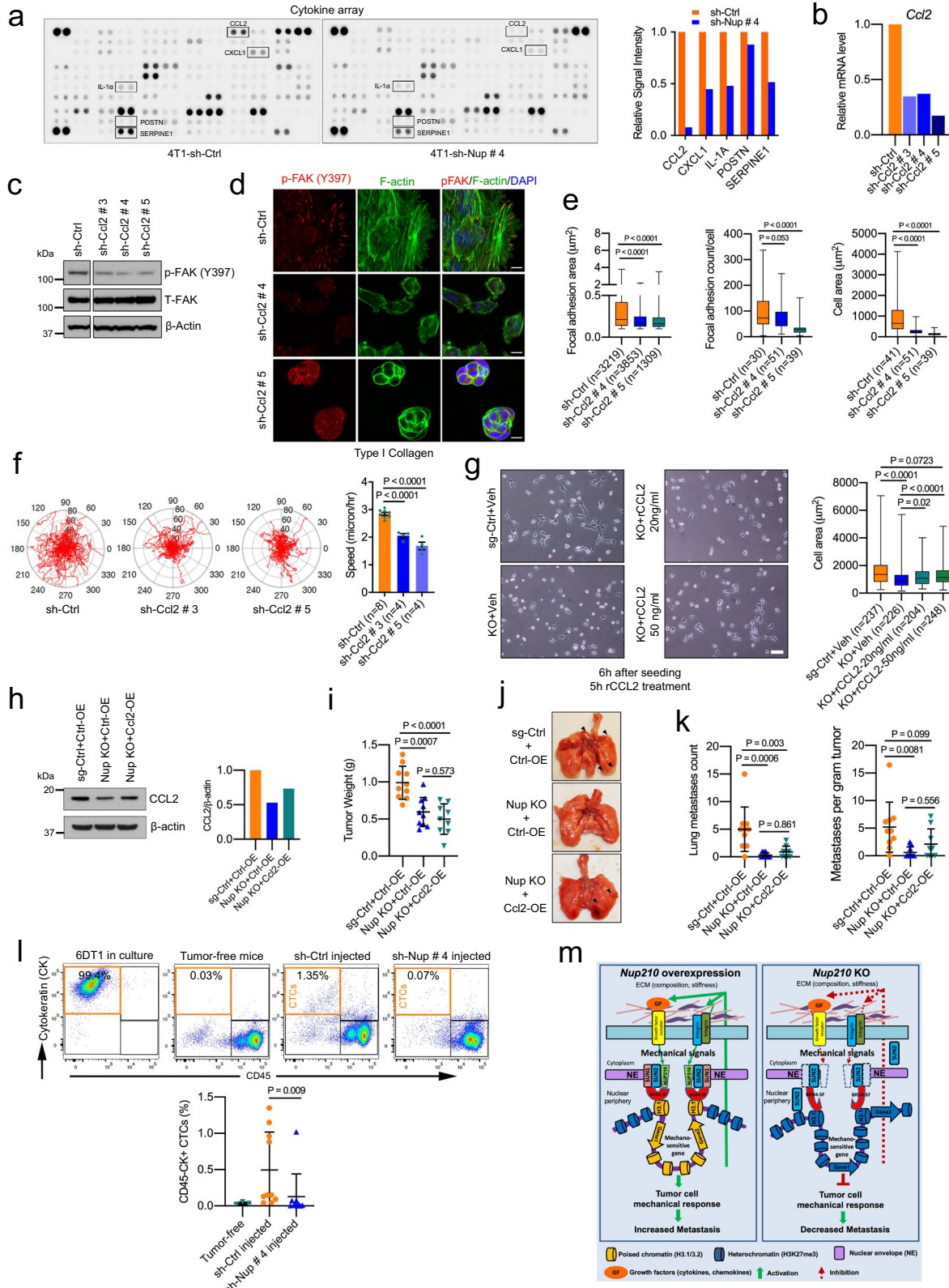

## Discussion

Apart from providing structural support and nucleocytoplasmic transport[57], the role of the NPC in developmental gene regulation has drawn much attention recently[58–61]. Although NUP210 has been implicated in muscle differentiation[28,62], cellular reprogramming[63], and T cell signaling[64,65], its function in human cancer was unknown. Here, we have identified *Nup210* as

a potential metastasis susceptibility gene for human ER+ breast cancer patients and demonstrated a previously unrecognized role of nuclear pore proteins in sensing the mechanical stress of the ECM.

The present study has important implications in the field of cellular mechanosensing. The nucleus is regarded as a sensor of mechanical stress where nuclear envelope-anchored, LINC

**Fig. 9 NUP210 regulates circulating tumor cells in mice. a** Cytokine analysis and quantification of boxed cytokines of supernatants from sh-Ctrl and *Nup210* KD 4T1 cells. **b** qRT-PCR analysis of *Ccl2* KD in 4T1 cells. **c** Western blot of p-FAK Y397 and T-FAK proteins in *Ccl2* KD 4T1 cells. **d** p-FAK (Y397) focal adhesion and F-actin distribution in sh-Ctrl and *Ccl2* KD 4T1 cells. Scale bar = 10 μm. **e** Quantification of focal adhesion area, count, and cell spreading in sh-Ctrl and *Ccl2* KD 4T1 cells. ANOVA with Tukey's multiple comparison correction, box represents 25th to 75th percentile, whiskers represent data range, and the horizontal line represents median. '*n*' equals the number of cells analyzed. **f** Cell migration tracks and quantification of sh-Ctrl and *Ccl2* KD 4T1 cells. Average cell speed analyzed by ANOVA with Dunnet's multiple comparison test, mean ± s.e.m. '*n*' equals the number of independent replicates. **g** Brightfield images and quantification of cell area of 4T1 *Nup210* KO cells treated with Ccl2. ANOVA with Tukey's multiple comparison correction, box represents 25th to 75th percentile, whiskers represent data range, and the horizontal line represents median. '*n*' equals the number of cells analyzed. **h** Western blot and signal quantification of *Ccl2* overexpression in *Nup210* KO 4T1 cells. **i** Tumor weight of mice injected with *Ccl2*-overexpressing *Nup210* KO 4T1 cells. ANOVA with Tukey's multiple comparison correction, mean ± s.d. For sg-Ctrl+Ctrl-OE and Nup KO + Ctrl-OE, *n* = 10 mice; for Nup KO + Ccl2-OE, *n* = 9 mice. (**j**) Representative lung images from the mice from (**i**). **k** Lung metastases count and metastasis normalized to tumor weight from the mice injected in panels **i** and **j**. ANOVA with Tukey's multiple comparison test, mean ± s.d. For sg-Ctrl+Ctrl-OE and Nup KO + Ctrl-OE, *n* = 10 mice; for Nup KO + Ccl2-OE, *n* = 9 mice. **l** (Top) Flow cytometry analysis of CD45−/CK+ circulating tumor cells (CTCs) and (bottom) quantification of CTCs in *Nup210* KD 6DT1 cell-injected animals. Kruskal–Wallis ANOVA with Dunn's multiple comparison test, mean ± s.d. Tumor-free, *n* = 3 mice; sh-Ctrl and sh-Nup # 4, *n* = 10 mice. **m** Proposed model of NUP210's metastatic role.

complex proteins are thought to serve as mechanotransducer to regulate gene expression[44]. Despite being embedded in the nuclear envelope, direct involvement of nuclear pore proteins in mechanosensation remains unclear. Previous studies have focused mainly on the translocation of mechanosensitive transcription factors to the nucleus in a nucleocytoplasmic transport-dependent manner[40,41,47]. In contrast, our study has identified a transport-independent role of nuclear pore components in regulating mechanosensitive, prometastatic genes through inter-action with LINC complex proteins and chromatin. However, we could not specifically rule out the causal effect of NUP210 loss due to the potential positive feedback loop between hetero-chromatin regulation and actin cytoskeletal tension. Further studies will be necessary to fully understand the mechanism of NUP210 in regulating cellular mechanosensation.

Intriguingly, many of the NUP210-regulated genes were immune molecules like chemokines and cytokines. This result has important implications since rapid transcriptional activation and secretion of immune molecules are required to induce an immune response[66]. Consistent with T cell defects in NUP210-deficient mice[64], the NPC might be essential for inducing an immune response. During metastasis, cancer cells may exploit this mechanism for robust transcription of these genes, which then promote their migration into distant organs in an autocrine fashion.

Mechanistically, we have demonstrated that the loss of NUP210 is associated with decreased peripheral localization of H3.1/3.2 and increased enrichment near heterochromatin foci. In support of this observation, we found increased accumulation of H3K27me3-marked heterochromatin in *Nup210* KO cells. These results suggest a role for NUP210 as a chromatin barrier insulator-binding protein. The NPC has been described as a heterochromatin barrier component of the nuclear lamina[67,68]. Loss of barrier insulator elements is associated with an increased spreading of heterochromatin marks to adjacent actively tran-scribing gene loci[69]. Since *Nup210* loss was associated with H3K27me3-marked heterochromatin spreading, NUP210 is probably acting as a barrier of H3K27me3-mediated hetero-chromatin spread to adjacent loci at the nuclear pore.

The potential enrichment of NUP210 at active enhancers and its association with chromatin reader BRD4-SF is consistent with the previous observation that the oncogenic properties of BRD4-SF are partly mediated through enhancer regulation[50]. Nuclear pores can be a scaffold of enhancer–promoter contact mediating transcriptional activation of poised genes[61]. It is therefore con-ceivable that in response to higher stiffness of the extracellular microenvironment, NUP210 can facilitate the interaction of LINC complex with BRD4-SF and histone H3.1/3.2-marked

enhancer regions to establish distal regulatory element interaction with poised promoters of mechanosensitive genes necessary for metastatic progression.

Treating *Nup210* KO cells with an EZH2 inhibitor (GSK126) could partially rescue the peripheral distribution of H3.1/3.2 and mechanosensitive gene expression. This has important clinical implications since EZH2 is a therapeutic target for multiple malignancies and some EZH2 inhibitors are currently in clinical trials[70,71]. In breast cancer, both tumor-promoting and inhibitory effects have been described for EZH2 in a context-specific manner[72]. Treating with an EZH2 inhibitor could adversely affect the patient outcome through activation of the mechanosensitive genes. In line with our observation, GSK126 treatment has been shown to exhibit adverse effects via inhibiting antitumor immunity[73] and promoting inflammation[74] in several preclinical models of cancer.

In conclusion, we have identified a previously unrecognized role for nuclear pore protein, NUP210, in mechanical sensing of aggressive metastatic cancer cells. Future studies should deter-mine whether blocking NUP210–chromatin interactions can be a therapeutic strategy to prevent metastasis.

## Methods

**Mouse strains.** Usage of animals described in this study was performed under the animal study protocol LCBG-004 approved by the National Cancer Institute (NCI) at Bethesda Animal Use and Care Committee. Animal euthanasia was performed by anesthesia using Avertin injection, followed by cervical dislocation. Female BALB/c (000651) and FVB/NJ (001800) mice were purchased from The Jackson Laboratory.

**Cell lines.** Mouse mammary tumor cell lines, 4T07, 4T1, 6DT1, and MVT1, were provided by Dr. Lalage Wakefield (NCI, NIH)[26]. These cell lines were grown in Dulbecco's modified Eagle's medium (DMEM) (Gibco) supplemented with 9% fetal bovine serum (FBS) (Gemini), 1% ʟ-glutamine (Gibco), and 1% penicillin–streptomycin (Gemini). Human breast cancer cell lines, MDA-MB-231 and MCF7, were provided by Dr. Jeffrey E. Green (NCI, NIH) and grown in DMEM with 9% FBS, 1% ʟ-glutamine, and 1% penicillin–streptomycin. Human 293FT cells were purchased from Thermo Fisher Scientific and grown on the medium mentioned above.

**BACh sequencing analysis.** Cells were expanded in DMEM at 37 °C to obtain 8–10 × 10⁷ cells/condition/each experiment. BACh analysis was performed as previously described with minor modifications[75]. Briefly, cells were collected by centrifugation, washed twice in ice-cold cellular wash buffer (20 mM Tris-HCl pH 7.5, 137 mM NaCl, 1 mM EDTA, 10 mM sodium butyrate, 10 mM sodium orthovanadate, 2 mM sodium fluoride, protease inhibitor cocktail (Roche) and resuspended in (40 million cells/ml) hypertonic lysis buffer (20 mM Tris-HCl pH 7.5, 2 mM EDTA, 1 mM EGTA, 0.5% glycerol, 20 mM sodium butyrate, 2 mM sodium orthovanadate, 4 mM sodium fluoride, and protease inhibitor cocktail). Cells were distributed in 500 μl aliquots in 1.5 ml tubes and followed by the addition of 500 μl of nuclease digestion buffer (40 mM Tris-HCl pH 8.0, 6 mM MgCl₂, 0.3% NP-40, and 1% glycerol) containing a 3-fold dilution (from 0.125 to 6 U/ml) of benzonase nuclease (Millipore). This was mixed gently and incubated

for 3 min at 37 °C. Reactions were terminated by the addition of EDTA (10 mM final concentration) and SDS (0.75% final concentration). Proteinase K was added to a final concentration of 0.5 mg/ml and incubated overnight at 45 °C. DNA fragments of 100–500 bp from a chromatin digestion were purified over sucrose gradients[76] and precipitated in 0.1 volume sodium acetate and 0.7 volume isopropanol.

**Sequencing and data analysis of benzonase-treated samples.** DNA was sequenced using either Illumina HiSeq2000 (TrueSeq V3 chemistry) or NextSeq500 (TrueSeq High output V2 chemistry) sequencers at the Advanced Technology Research Facility, National Cancer Institute (NCI-Frederick, MD, USA). The sequence reads were generated as either 50 or 75 mer (trimmed to 50 mer by Trimmomatic software before alignment), and tags are then aligned to the UCSC mm9 reference genome assembly using Eland or Bowtie2. All the samples were of good quality with over 94% of the bases having Q30 or above with 25–50 million raw reads per sample. Regions of enriched tags termed hotspots have been called using DNase2Hotspots algorithm[77] with an FDR of 0% with minor updates. Tag density values were normalized to 10 million reads. For comparison of BACh profiles across multiple cell lines, percent reference peak coverage (PRC) measure was proposed and employed to ensure compatible levels of digestion by benzonase in multiple cell lines[78]. To calibrate PRC, commonly represented hotspot sites were identified as reference peaks based on the mouse (mm9) DNaseq data from ENCODE. ENCODE narrowPeak definition files (DNaseI Hypersensitivity by Digital DNaseI from ENCODE/University of Washington) for a total of 133 samples were downloaded from the UCSC golden path website: http://hgdownload.cse.ucsc.edu/goldenPath/mm9/encodeDCC/wgEncodeUwDnase/. A total of 8587 peaks were found to be present in all 133 samples. By assuming that the most commonly accessible sites may also be present in our optimally digested samples, PRC was obtained as a fraction of reference peaks presented in each sample. Our pooled samples with a PRC of 90% or greater indicated that all the 8587 sites were present in the sample. Two biological replicates with acceptable PRC were selected and pooled for each cell line, and each pooled dataset showed over 90% PRC. When samples showed bimodality in their tag density distribution due to elevated noise, we dropped hotspots that belonged to a group with low tag density and low PRC by applying a threshold to maximum hotspot tag density values.

**Computing environment.** All computations at NCI were performed on the NIH helix/biowulf system, documentation of which is available at https://helix.nih.gov. We used the R computing environment, Perl scripts, Bedtools, and UCSC liftOver for most of the analyses.

**Identification of polymorphic BACh sites.** The workflow consisted of the following: (1) the BACh data were filtered for the regions overlapping with polymorphic sites. Since the BACh data were generated in Genome Build mm9, we used UCSC mm9 snp128 data to restrict the BACh sites. (2) Variant called format files were filtered to retain the SNPs that overlap with the BACh present in the 4T1 and 4T07 cell lines. (3) SNPs were removed in the BACh that is present in the mouse FVB/NJ strain.

**Long read ChIA-PET sequencing and data processing.** Long read ChIA-PET was performed using Tn5 transposase to tag DNA for long tag sequencing by Illumina NextSeq. ChIA-PET data was processed by a customized ChIA-PET data processing pipeline[22]. A detailed protocol can be found in Li et al.[79].

**Cloning.** A mouse *Nup210* full-length complementary DNA (cDNA) (NM_018815) encoding vector (pCMV6-Nup210-Myc) was purchased from Origene Technologies. This vector was digested with SalI-HF (New England Biolabs) and EcoRV-HF (New England Biolabs) restriction enzymes to obtain the NUP210 protein-encoding region. This NUP210 insert was then cloned into the Gateway entry vector pENTR1A using the Quick Ligation Kit (New England Biolabs). pENTR1A no ccdB plasmid (Addgene plasmid # 17398) was received by Dr. Marian Darkin (NCI) as a gift from Dr. Eric Campeau. The NUP210-encoding region from the pCMV6-NUP210-Myc vector was cloned into pENTR1A vector through digestion with SalI and EcoRV. pENTR1A-NUP210 entry vector was used to transfer the NUP210-encoding region into the lentiviral destination vector pDEST-658 (received as a gift from Dr. Dominic Esposito, NIH) along with a mouse *Pol2* promoter entry vector using the Gateway LR clonase reaction (Thermo Fisher Scientific). The integrity of the final NUP210-encoding vector was verified through DNA-sequencing.

The 3× Flag-histone H3.1 plasmid was received as a gift from Dr. Jing Huang (NCI, NIH). The coding region of H3.1 along with Flag-Tag was amplified by PCR using KOD Hot Start DNA Polymerase (Millipore) and cloned into gateway entry vector pENTR/D-TOPO vector (Thermo Fisher Scientific). Entry vectors containing Flag-H3.1 were then cloned into gateway lentiviral destination vector pDest-659 (gift from Dr. Dominic Esposito Lab) along with mPol2 promoter and C-terminal V5-tag entry vectors using the gateway LR clonase reaction (Thermo Fisher Scientific).

The pGL4.23 [*luc2*/minP] promoter-luciferase vector (Promega, Catalog # E8411) was received as a gift from Dr. Jing Huang (NCI, NIH). A 550 bp region from the mouse (FVB/NJ and BALB/cJ) *Nup210* promoter covering the polymorphic sites was amplified by PCR using KOD Hot Start DNA Polymerase (Millipore) and digested with KpnI (New England Biolabs) and XhoI (New England Biolabs) restriction enzymes. Gel-purified PCR product was then cloned into the pGL4.23 vector using T4 DNA ligase (New England Biolabs). Finally, the DNA sequence of the clones was confirmed through DNA-sequencing.

pMXs-puro-EGFP-FAK plasmid was obtained as a gift from Dr. Noboru Mizushima (University of Tokyo) (Addgene # 38194). cDNA encoding mouse FAK (Ptk2) without EGFP was PCR amplified from this plasmid and subcloned into pENTR/D-TOPO entry vector (Thermo Fisher Scientific). Entry vector is then further cloned into lentiviral destination vector pDest-659 along with mPol2 promoter and C-terminal myc-tag entry vectors using the method described above. cDNA encoding mouse *Ccl2* was PCR amplified from 4T1 cells and subcloned into pENTR/D-TOPO entry vector. Ccl2 entry vector is then further cloned into lentiviral destination vector pDest-659 along with mPol2 promoter and C-terminal myc-tag entry vectors using the above method. The sequence of the final clones were confirmed through Sanger sequencing.

**Lentivirus production and generation of stable cell lines.** All of the TRC lentiviral shRNA vectors were purchased from Dharmacon. shRNAs targeting the mouse *Nup210* gene, sh-Nup # 1 (TRCN0000101935: TAACTATCACAGTAAG AAGGC) and sh-Nup # 4 (TRCN0000101938: TTCAGTTGCTTATCTGTCAGC), were used for *Nup210* knockdown in all of the mouse cell lines. For mouse *Ctcf* knockdown, sh-Ctcf # 1 (TRCN0000039022: TAAGGTGTGACATATCATCGG) and sh-Ctcf # 2 (TRCN0000039023: ATCTTCGACCTGAATGATGGC) were used. For mouse *Ccl2* knockdown, sh-Ccl2 # 3 (TRCN0000034471: TTACGGGTCAACTTCACATTC), sh-Ccl2 # 4 (TRCN0000034472: TTGCTGGTGAATGAGTAGCAG), and sh-Ccl2 # 5 (TRCN0000034473: AATGTATGTCTGGACCCATTC were used. For NUP210 knockdown in human cell lines, an shRNA targeting the human *NUP210* gene (TRCN0000156619: AAATGAGCTAATGGGCAGAGC) was used.

For lentivirus production, shRNA-containing plasmids and packaging plasmids, psPAX2 (Addgene plasmid # 12260) and envelope plasmid pMD2.G (Addgene plasmid # 12259) (both were a gift from the Trono lab), were transfected into the human 293FT (Thermo Fisher Scientific) cell line using X-tremeGENE 9 DNA transfection reagent (Roche). At 48 h after transfection, culture supernatant containing lentivirus was harvested, filtered through a 0.45 µm filter (Millipore), and then used for transduction of mouse and human breast cancer cell lines. shRNAs stably integrated into mouse and human cells were selected with 10 and 2 µg/ml puromycin (Sigma), respectively. For the selection of NUP210-overexpressing cells, 10 µg/ml of blasticidin (Gibco) was used.

**CRISPR/Cas9-mediated knockout of the mouse *Nup210* gene.** For CRISPR/Cas9-mediated knockout of mouse *Nup210* in the 4T1 cell line, single-guide RNA (sgRNA) targeting *Nup210* exon 5 (sgRNA sequence: GCGACACCATCCTAG TGTCT) was designed using the GPP Web Portal available at the Broad Institute (https://portals.broadinstitute.org/gpp/public/analysis-tools/sgrna-design). sgRNA was then cloned into the lentiGuide-puro (Addgene plasmid # 52963, a gift from the Feng Zhang lab) vector. Nontargeting control sgRNA cloned into the lentiGuide-puro vector (sgRNA sequence: CCATATCGGGGCGAGACATG) was kind of a gift from Dr. Ji Luo (NCI, NIH). Lentiviral particles were prepared as described above for shRNA lentiviruses. 4T1 cells stably expressing the sgRNAs were generated through lentiviral transduction and selection with 10 µg/ml puromycin. For transient expression of Cas9 in the sgRNA-stable 4T1 cells, an adenoviral Cas9-encoding viral particle containing the GFP reporter (Vector Biolab) was used at 25 MOI (multiplicity of infection) 96 h after transfection of the Cas9 vector, GFP-positive 4T1 cells were FACS sorted and single cells were isolated for clonal expansion. *Nup210* mutation was confirmed through DNA-sequencing and knockout was verified through western blot. Before performing functional assays with *Nup210* KO cells, both sg-Ctrl and KO cells were passaged at least four times for 2 weeks to eliminate residual Cas9-GFP signal within 4T1 cells, which was verified using a fluorescence microscope.

**CRISPR/Cas9 D10A-mediated deletion of CTCF-binding site on *Nup210* promoter.** For the deletion of polymorphic CTCF-binding region on mouse *Nup210* promoter, Cas9 D10A double-nicking strategy was used. Sense and antisense sgRNAs spanning the CTCF-binding site were designed using the above link of the GPP web portal of Broad Institute. All-in-One plasmid encoding dual U6 promoter-driven sgRNAs and EGFP-coupled Cas9 D10A plasmid (AIO-GFP) was a gift from Dr. Steve Jackson (The Wellcome Trust Sanger Institute, UK) (Addgene # 74119). sgRNAs were cloned into AIO-GFP plasmid using the method described in the paper[80]. Briefly, the AIO-GFP plasmid was digested with BbsI restriction enzyme, dephosphorylated using calf intestinal phosphatase, and gel purified. sgRNA oligonucleotide pairs were purchased from Integrated DNA Technologies, annealed, and phosphorylated using T4 polynucleotide kinase (NEB). First, sgRNA was cloned into BbsI site, and second, sgRNA was cloned into BsaI site. The DNA sequence of the final clone was verified via Sanger sequencing using the primer (5′-

CTTGATGTACTGCCAAGTGGGC-3′). sgRNA plasmid was then transfected into 4T1 cells using Nanojuice transfection reagent (Millipore). After 48 h, GFP-positive cells were sorted using flow cytometry and single cells were plated on a 96-well plate. Cas9 D10A-edited clones were identified using the PCR and DNA Sanger sequencing.

**Spontaneous metastasis assay in mice**. Six- to eight-week-old female BALB/c and FVB/NJ mice were purchased from The Jackson Laboratory. For orthotopic transplantation of gene knockdown or overexpressing cells, 100,000 cells were injected into the fourth mammary fat pad of mice. At 28–30 days after injection, mice were euthanized and primary tumors were resected, weighed, and surface lung metastases were counted. For all the mouse experiments, the ethics committee-permitted maximum tumor size/burden was not exceeded.

**Protein nucleocytoplasmic transport assay**. 4T1 *Nup210* knockdown cells were grown on glass coverslips in a 2-well Lab-Tek chambered glass coverslip (Thermo Fisher Scientific) at a seeding density of 20,000 cells per well. At 24 h after seeding, cells were transfected with 1 μg of nucleocytoplasmic transport reporter (NLS-tdTomato-NES) plasmid (received as a gift from Dr. Martin W. Hetzer, Salk Institute) using Novagen Nanojuice Transfection Reagent (Millipore-Sigma). At 4 h after transfection, the medium was replaced with a fresh medium and cells were grown for 24 h. Cells were then treated with 20 nM leptomycin B (Cell Signaling Technology) for 6 h and then fixed with methanol for immunostaining with an antibody against NPC proteins (clone mAb414, Abcam) following the regular immunofluorescence protocol described below. Nuclear and cytoplasmic expression of tdTomato was observed using confocal microscopy.

**Protein interaction analysis through liquid chromatography-mass spectrometry (LC-MS)**. NUP210-Myc-overexpressing 4T1 cells were seeded onto 15 cm tissue culture dishes at a seeding density of $2.5 \times 10^6$ cells per dish. After 48 h of incubation, cells were harvested and nuclear protein complex lysates were prepared using the Nuclear Complex Co-IP Kit (Active Motif). Co-IP with two biological replicates was performed according to the manufacturer's instructions. Briefly, 500 μg of nuclear protein lysates were incubated with 2 μg of either Myc-Tag antibody (Cell Signaling Technology) or an endogenous NUP210-specific antibody (Bethyl Laboratories). After overnight incubation on a rotator at 4 °C, 25 μg of Dynabeads Protein G (Invitrogen) were added to the protein lysate–antibody complexes. After 30 min of incubation with beads, antibody–bead–protein complexes were isolated using a magnetic stand. Beads were then washed three times and then dissolved in 25 mM ammonium bicarbonate pH 8.0 (Sigma). Samples were then subjected to LC-MS analysis.

**Western blot**. Whole-cell protein lysate was prepared using lysis buffer (20 mM Tris-HCl pH 8.0, 400 mM NaCl, 5 mM EDTA, 1 mM EGTA, 10 mM NaF, 1 mM sodium pyrophosphate, 1% Triton X-100, 10% glycerol, protease and phosphatase inhibitor cocktail). Nuclear and cytoplasmic protein lysates were prepared using the Nuclear Extract Kit (Active Motif) or Nuclear Complex Co-IP Kit (Active Motif). Protein concentration was measured using the Pierce BCA Protein Assay Kit (Thermo Fisher Scientific). Twenty-five micrograms of protein lysates was mixed with 4× NuPAGE LDS sample buffer (Invitrogen) and 10× NuPAGE Sample Reducing Agent (Invitrogen). Samples were then boiled at 95 °C for 5 min and resolved on NuPAGE 3–8% Tris-acetate, NuPAGE 4–12% Bis-Tris, or Novex 4–20% Tris-glycine protein gels (Thermo Fisher Scientific) with appropriate running buffer. Protein was transferred onto a PVDF membrane (Millipore) and the membrane was blocked with blocking buffer (TBST + 5% non-fat dry milk) for 1 h. Membranes were then incubated with appropriate primary antibodies overnight. After washing with TBST, membranes were incubated with secondary antibodies for 1 h. Finally, the signal was developed on X-ray film using the Amersham ECL Western Blotting Detection Reagent (GE Healthcare). Densitometry quantification of western blot was performed using the FIJI software[81].

Primary antibodies and their dilutions were as follows: NUP210 (1:500; Bethyl Laboratories), β-actin (1:10,000; Abcam), Lamin B1 (1:5000; Abcam), Lamin A/C (1:1000; Abcam), Myc-Tag (1:1000; Cell Signaling Technology), rabbit V5-Tag (1:1000; Cell Signaling Technology), mouse V5-Tag (1:1000; Cell Signaling Technology), RRP1B (1:1000; Millipore-Sigma), SUN1 (1:500; Abcam), SUN2 (1:1000; Abcam), FAK (1:5000; Abcam), p-FAK(Y397) (1:5000; Abcam), p-FAK (Y397) (1:5000; Thermo Fisher Scientific), CCL2 (1:1000; Proteintech), H3.1/3.2 (1:1000; Active Motif), H3K27me3 (1:5000; Cell Signaling Technology), H3K9me3 (1:5000; Abcam), SUV39H1 (1:1000, Cell Signaling Technology), EZH2 (1:1000; Cell Signaling Technology), and SUZ12 (1:1000; Cell Signaling Technology). Anti-mouse horse radish peroxidase (HRP) secondary antibody (GE Healthcare) was used at 1:10,000 dilution and anti-rabbit HRP secondary antibody (Cell Signaling Technology) was used at 1:3000 dilution.

**Coimmunoprecipitation**. Co-immunoprecipitiation was performed using the Nuclear Complex Co-IP Kit (Active Motif). 4T1 cells were seeded onto 15 cm tissue culture dishes at a seeding density of $5 \times 10^6$ cells per dish. After 48 h of incubation, cells were harvested and nuclear lysates were prepared. A total of 200–500 μg of nuclear lysates were incubated with 2 μg of specific antibodies and 50 μg of

Dynabeads Protein G (Invitrogen). After overnight incubation on a rotator at 4 °C, immune complexes were isolated using a magnetic stand. Beads were then washed three times, resuspended in 2× NuPAGE LDS sample buffer (Invitrogen), and incubated at 95 °C heat block for 5 min. Samples were loaded onto NuPAGE protein gels and the standard western blot protocol was followed as described above.

For co-IP in 293FT cells, $2.5 \times 10^6$ cells were seeded into 15 cm tissue culture dishes. Flag-H3.1 (5 μg) and NUP210-Myc (5 μg) plasmids were cotransfected using X-tremeGENE 9 transfection reagent. At 48 h after transfection, cells were harvested and nuclear protein lysates were prepared using the Nuclear Complex Co-IP Kit (Active motif). Co-IP was followed as described above.

**Immunofluorescence and confocal microscopy**. Immunofluorescence analysis was performed as described previously[28]. Briefly, cells were grown on 4- or 8-well polymer coverslips (Ibidi) at a seeding density of 40,000 or 20,000 cells per well, respectively. After 24 h of incubation, cells were fixed with −20 °C methanol for 2 min and permeabilized with phosphate-buffered saline (PBS) containing 1% Triton X-100 for 1 min. Fixed cells were then blocked with immunofluorescence buffer (1× PBS, 10 mg/ml bovine serum albumin (BSA), 0.02% SDS, and 0.1% Triton X-100) for 30 min. Cells were then incubated with primary antibodies diluted in immunofluorescence buffer overnight at 4 °C. After washing the cells with immunofluorescence buffer three times for 10 min per wash, cells were incubated with Alexa Fluor-conjugated secondary antibodies for 1 h at room temperature. Cells were washed three times with immunofluorescence buffer and then incubated with 1 μg/ml DAPI for 10 min to stain the nucleus. After washing the cells with PBS three times, slides were kept at 4 °C until subjected to confocal microscopy. In cases of drug treatment followed by super-resolution microscopy, cells were treated with 1 μM JQ1, 5 μM GSKJ4, and 0.5 μM CytoD for 24 h. Images were acquired using either a Zeiss LSM 780 confocal microscope (×63 plan-apochromat NA 1.4 oil-immersion objective lens, 0.09 μm X–Y pixel size and 1.0 μm optical slice thickness), a Zeiss LSM 880 Airyscan super-resolution microscope (Airyscan detector, ×63 plan-apochromat NA 1.4 oil-immersion objective lens and 0.05 μm X–Y pixel size), a Zeiss LSM 780 Elyra with SIM module or a Nikon SoRa spinning disk super-resolution microscope (Yokogawa SoRa spinning disk unit, ×60 plan-apochromat NA 1.49 oil-immersion objective lens, Photometrics BSI sCMOS camera, and 0.027 μm X–Y pixel size). Airyscan images were processed using the Airyscan processing algorithm in the Zeiss ZEN Black (v.2.3) software, whereas the Nikon SoRa images were deconvolved using a constrained iterative restoration algorithm in the Nikon NIS Elements (v5.11) software. Tetraspeck 0.2 μm beads (Invitrogen) were imaged with the same microscope parameters and used for channel alignment. Further image processing was done in Zeiss ZEN Blue V2 software.

Primary antibodies used for immunofluorescence were as follows: p-FAK (Y397) (1:100; Abcam), H3.1/3.2 (1:1000; Active Motif), mouse Myc-Tag (1:500; Cell Signaling Technology), rabbit Myc-Tag (1:100; Cell Signaling Technology), SUN1 (1:100; Abcam), SUN2 (1:100; Millipore), p-MLC2-S19 (1:100; Cell Signaling Technology), p-FAK Y397 (1:500 for tissue IF; Thermo Fisher Scientific), H3K4me3 (1:1000; Millipore), rabbit H3K27me3 (1:1000; Cell Signaling Technology), mouse H3K27me3 (1:250; Abcam), H3K27Ac (1:500; Cell Signaling Technology), H3K9me3 (1:1000; Abcam), Nucleolin (1:500; Abcam), Lamin B1 (1:500; Abcam), Lamin A/C (1:500; Abcam), and NPC antibody (clone mAb414, 1:1000; Abcam). Secondary antibodies used were mouse Alexa Fluor 488 (1:200; Invitrogen), Phalloidin Alexa Fluor 488 (1:250; Invitrogen), rabbit Alexa Fluor 488 (1:200; Invitrogen), rabbit Alexa Fluor 568 (1:200; Invitrogen), rabbit Alexa Fluor 594 (1:200; Invitrogen), and mouse Alexa Fluor 594 (1:200).

**Immunofluorescence on tumor tissue**. Frozen tumor tissue sections were fixed in 4% paraformaldehyde in PBS for 10 min at room temperature. Slides were then washed with PBS three times and permeabilized with 0.5% Triton X-100 in PBS for 3 min. After washing with 1× PBS three times, slides were then incubated with a blocking solution (10% BSA in PBS) for 1 h. Primary antibodies were diluted in blocking solution, added to the tissue sections, and incubated overnight at 4 °C. Slides were washed three times with PBS + 0.1% Tween-20 (PBST) for 5 min and incubated with secondary antibodies in blocking buffer for 1 h at room temperature. Sections were then washed with 1× PBST three times for 5 min and incubated with DAPI solution (1 μg/ml) for 10 min at room temperature. Slides were washed with 1× PBS three times for 5 min and then mounted with Prolong Glass Antifade mountant (Invitrogen). Confocal Z-stack images were acquired using Nikon SoRA spinning disk microscope. Dilution of the primary antibodies are as follows: rabbit p-FAK Y397 (1:500; Thermo Fisher Scientific), mouse Alexa Fluor 488-conjugated pan-cytokeratin (1:100; Cell Signaling Technology). Dilution of the secondary antibodies are as follows: anti-rabbit Alexa Fluor 568 antibody (1:200; Invitrogen).

**FRAP assay**. 4T1 cells were grown on 35 mm glass-bottom dishes (MatTek). After overnight incubation, cells were transfected with 0.5 μg POM121-GFP plasmid (Genecopoeia) using Nanojuice transfection reagent. At 24 h after transfection, cells were subjected to FRAP analysis in Zeiss 880 confocal microscope. The percent mobile fraction of POM121-GFP in each condition was calculated using the FRAP module fit formula option of the ZEN software (Zeiss).

**Cell cycle analysis**. Cell cycle analysis was performed using the Click-iT EdU (5-ethynyl-2′-deoxyuridine) Alexa Fluor 488 Flow Cytometry Assay Kit (Thermo Fisher Scientific) and FxCycle Violet Stain (Thermo Fisher Scientific) according to the manufacturer's instructions. Briefly, 4T1 cells were seeded onto 15 cm tissue culture dishes at a seeding density of $3 \times 10^6$ cells per dish. After 24 h, cells were pulsed with Click-iT EdU for 1 h. After harvesting the cells through trypsinization, cells were fixed with Click-iT fixative and permeabilized with a saponin-based permeabilization agent. The Click-iT reaction was then performed for 30 min at room temperature. Cells were then washed with wash buffer and stained for DNA content analysis with FxCycle Violet, a DNA-selective dye. Finally, cell cycle analysis was performed using a BD FACSCanto II flow cytometer (BD Bioscience). Data were analyzed using the FlowJo V10 (FlowJo, LLC) software.

**ChIP-seq analysis**. ChIP was carried out using the ChIP-IT Express Enzymatic Chromatin Immunoprecipitation Kit (Active Motif) according to the manufacturer's instructions. Briefly, $5 \times 10^6$ 4T1 cells were seeded onto 15 cm tissue culture dishes. After 48 h of incubation, cells were fixed with 1% formaldehyde for 10 min at room temperature. Cells were washed with ice-cold PBS and formaldehyde cross-linking was quenched using glycine stop-fix solution. Cells were harvested through scraping and pelleted by centrifugation at 2500 r.p.m. for 10 min at 4 °C. After cell lysis with ice-cold lysis buffer and a Dounce homogenizer, chromatin was sheared using an enzymatic shearing cocktail for 12 min at 37 °C. The shearing reaction was stopped with 0.5 M EDTA and the chromatin was separated through centrifugation at 15,000 r.p.m. for 10 min at 4 °C. Sixty to seventy micrograms of chromatin was used with 25 μl of Protein G magnetic beads and 2 μg of specific antibodies for each ChIP reaction. The following antibodies were used for the ChIP reactions: anti-rabbit NUP210 (Bethyl Laboratories), anti-rabbit H3K27me3 (Cell Signaling Technology), anti-rabbit H3K4me3 (Millipore), and anti-mouse NPC antibody (clone mAb414, Abcam). After incubating the reaction mixture overnight at 4 °C on a rotator, the beads were washed with ChIP buffers and DNA was eluted with elution buffer. DNA was then reverse-cross linked, proteinase K-treated, and DNA quality was assessed using an Agilent Bioanalyzer before ChIP-seq library preparation. The library was prepared using a TruSeq ChIP Library Preparation Kit (Illumina) and pooled samples were sequenced on the NextSeq platform.

ChIP-seq data were analyzed using the Sicer algorithm[82] with default parameters. For narrow peaks such as H3K4me3 enrichment, a window size of 200 was used. For broad peaks such as H3K27me3 and NUP210, a window size of 1000 was used. The ChIP-seq plot profile was generated using deepTools2[83]. For ChIP-seq peak annotation, the ChIPSeeker[84] Bioconductor package was used. BigWig files were displayed using the Integrative Genomics Viewer[85].

**Publicly available ChIP-seq and Hi-C data analysis**. H3K27Ac ChIP-seq data from mouse mammary luminal cells were downloaded from Gene Expression Omnibus (GEO) (accession no.: GSE116384)[26]. CTCF ChIP-seq data from mouse mammary epithelial cells were obtained from GEO (accession no.: GSE92587, sample: GSM2433042)[20]. CTCF and H3K27Ac ChIP-seq data from the MCF7 human breast cancer cell line were obtained from GEO (accession no.: GSE130852). Publicly available Hi-C data from mESCs were derived from Bonev et al.[37] and displayed using the 3D-Genome Browser[38].

**RNA isolation and quantitative reverse transcriptase real-time PCR (qRT-PCR)**. Cells were directly lysed on cell cultures plate with 1 ml TriPure Isolation Reagent (Sigma). After adding 200 μl chloroform and centrifuging at 14,000 r.p.m. for 15 min at 4 °C, the upper aqueous layer containing the RNA was transferred to a new tube. To precipitate RNA, 500 μl of isopropanol was added to each tube, which was then vortexed and incubated at −20 °C for 1 h. The RNA was further purified using the RNeasy Mini Kit (Qiagen) with on-column DNase (Qiagen) digestion according to the manufacturer's instructions. Two micrograms of total RNA was used for cDNA preparation using the iScript cDNA Synthesis Kit (Bio-Rad). A 1/10 dilution of cDNA was used for qRT-PCR analysis using the FastStart Universal SYBR Green Master Mix (Roche). Sequences of the primers used are listed in Supplementary Table 1.

**ChIP qPCR analysis**. Chromatin was isolated from 4T1 and 6DT1 cells using the ChIP-IT Express Enzymatic Kit (Active Motif) according to the manufacturer's instructions as mentioned above. Ten microliters of normal rabbit IgG (Cell Signaling Technology), 10 μl of rabbit CTCF (Cell Signaling Technology), and 10 μl of rabbit H3K27Ac (Cell Signaling Technology) antibodies were used for the immunoprecipitation reaction. ChIP DNA was then subjected to qPCR analysis. The percent (%) input method was used for the calculation of CTCF and H3K27Ac enrichment.

**RNA-seq analysis**. 4T1 cells were seeded onto 6 cm tissue culture dishes at a seeding density of $4 \times 10^5$ cells per dish. After 48 h of incubation, total RNA was isolated using the protocol described above. On-column DNase treatment was performed to eliminate DNA contamination. Library preparation was performed using the TruSeq Stranded mRNA Library Prep Kit (Illumina) and pooled samples were sequenced on a HiSeq2500 with TruSeq V4 chemistry (Illumina). Differential

gene expression analysis from RNA-seq data was performed using the Partek Flow software. GO enrichment analysis of differentially expressed genes was performed using the PANTHER classification system[86].

**Promoter-luciferase assay**. The 24-well tissue culture plates were coated overnight with 100 μg/ml type I collagen. 293FT cells were seeded in an antibiotic-free medium at a seeding density of 75,000 cells per well of 24-well plate. Two hundred and fifty nanograms of pGL4.23 Nup210 luciferase promoter vectors and 25 ng of pRL-TK Renilla luciferase vectors were cotransfected using NanoJuice transfection reagent (Millipore). At 24 h after transfection, the luciferase assay was performed using Dual-Luciferase Reporter Assay System Kit (Promega) according to the manufacturer's instructions. Luciferase activity was measured using a GloMax 96 Microplate Luminometer (Promega). Firefly luciferase activity was normalized with Renilla luciferase activity. Eight biological replicates were used per condition.

**Cytokine array**. 4T1 Nup210 knockdown cells were cultured in low serum condition (1% FBS) in 6-well plates at a seeding density of 500,000 cells per well. At 24 h after incubation, cell culture supernatants were harvested and centrifuged to remove the dead cells. One milliliter of supernatant was used for cytokine profiling using the Proteome Profiler Mouse XL Cytokine Array (R & D Biosystems) according to the manufacturer's instructions. Chemiluminescent signals were quantified using the ImageJ (NIH) software.

**3D-DNA FISH analysis**. For DNA FISH analysis, cells were grown on Lab-Tek chamber slides. Cells were briefly washed with PBS and fixed with 4% paraformaldehyde. Cells were permeabilized with 0.2% Triton X-100 in PBS and then hybridized with a labeled probe. For probe generation, BAC clones were purchased from CHORI (https://bacpacresources.org/). The following BAC clones were used for the experiment: Cxcl region: clone RP23-374O6; Postn region: clone RP23-480C1; Ccl2 region: clone RP23-99N1, and Itgb2 region: clone RP23-166E21. All the clones were expanded and DNA was isolated using the FosmidMax DNA Purification Kit (Epicentre). DNA from each clone was labeled through nick translation with either the Atto550 NT Labeling Kit (Jena Bioscience) or the Digoxigenin NT Labeling Kit (Jena Bioscience). Hybridization was carried out in a humidified chamber at 37 °C for 16 h. Post-hybridization rapid wash was carried out with 0.4× SSC at 72 °C for 4 min. Digoxigenin was detected with a DyLight 594-Labeled Anti-Digoxigenin/Digoxin (DIG) antibody (Vector Laboratories). The slides were stained with DAPI and Z-stack images were captured using a Zeiss LSM 880 Airyscan microscope.

**Response to ECM stiffness assay**. To test the ECM stiffness effect on tumor cells, polyacrylamide hydrogel-bound cell culture plates with different elastic moduli was used as a mimic of ECM stiffness. 6-well plates with 0.2 kPa (soft) or 12 kPa (stiff) elastic moduli were purchased from Matrigen. Each well of the plate was coated with 20 μg/ml fibronectin (Sigma) in PBS for 1 h at 37 °C. After washing the well with PBS, 4T1 cells were seeded on top of the soft or stiff matrix at a seeding density of 100,000 cells per well. At 48 h after incubation, 4T1 cells were harvested for RNA or protein isolation.

**Focal adhesion and cell spreading assay**. For focal adhesion immunostaining, Lab-Tek 2-well glass chamber slides (Thermo Fisher Scientific) were coated with 50 μg/ml of collagen type I (Gibco) or 20 μg/ml fibronectin. Cells were seeded at a seeding density of 20,000–50,000 cells per well. After 6 h or overnight incubation, cells were fixed in 4% paraformaldehyde for 20 min. Cells were then washed with PBS, permeabilized with PBS + 0.1% Triton X-100 for 5 min, and blocked with PBS + 5% normal goat serum. Cells were incubated with phospho-FAK (Y397) antibody for 1 h. After washing with PBS, cells were incubated with Alexa Fluor 594-conjugated rabbit secondary antibody and Alexa Fluor 488-conjugated phalloidin (F-actin staining) for 1 h. The slides were then washed with PBS and mounted using DAPI (1 μg/ml) or VECTASHIELD with DAPI Mounting Medium (Vector Laboratories). Images were captured using a Zeiss 780/880 confocal microscope.

To check the effect of recombinant CCL2 on cell spreading of Nup210 KO cells, 50,000 4T1 sg-Ctrl and Nup210 KO cells were seeded onto Lab-Tek glass coverslip coated with type I collagen (50 μg/ml). Cells were allowed to attach on a coverslip for 1 h and then cells were incubated with murine recombinant CCL2 (Peprotech) for 5 h. Images were captured using a brightfield microscope and cell area was quantified using ImageJ (NIH) software.

**Actin polymerization inhibition using cytochalasin D**. For the cytochalasin D treatment of 4T1 Nup210 knockdown cells, 20,000 cells were seeded onto 4-well μ-Slides with polymer coverslips (Ibidi). After 24 h, cells were treated with 1 μM CytoD, a potent inhibitor actin polymerization, for 2 h. After the treatment, cells were washed with complete medium three times and incubated for another 2 h in a 37 °C $CO_2$ incubator for the recovery of actin polymerization. Cells were then fixed with 4% paraformaldehyde for 30 min and stained with Alexa Fluor 488-conjugated phalloidin for immunofluorescence imaging. Nuclei were stained with 1 μg/ml Hoechst 33342 (Thermo Fisher).

For the western blot assay, $2.5 \times 10^6$ 4T1 cells were seeded onto 15 cm cell culture dishes. After 24 h, cells were treated with 0.5 μM CytoD and incubated 24 h before harvesting the cells for protein isolation. For RNA isolation, $3 \times 10^5$ cells were seeded onto 6 cm dishes. After 24 h later, cells were treated with 0.5 μM CytoD and incubated 24 h before lysing the cells for RNA isolation.

**Histone-modifying enzyme inhibitor treatment**. For immunofluorescence microscopy, 4T1 sg-Ctrl and *Nup210* KO cells were seeded onto 4-well μ-Slides at a seeding density of 25,000 cells per well. After 24 h, cells were treated with dimethyl sulfoxide, 5 μM EZH2 inhibitor GSK126 (Selleckchem), or 5 μM H3K27me3 demethylase inhibitor GSKJ4 (Selleckchem) for another 24 h. Cells were then fixed with −20 °C methanol for 2 min. Histone H3.1/3.2 and H3K27me3 antibodies were used for immunofluorescence staining according to the protocol described above[28].

For the qRT-PCR analysis, $2 \times 10^5$ 4T1 *Nup210* KO cells were seeded onto 6 cm dishes. After 24 h, 5 μM GSK126 or 5 μM GSKJ4 was applied to the cells that were then incubated for another 24–48 h. Cells were then lysed, RNA was isolated, and qRT-PCR was performed as described above.

**Live-cell imaging assay**. For the automated random cell migration assay, 4T1 cells were seeded onto 96-well polystyrene microplates (Corning) at a seeding density of 2000 cells per well so that the cell density remained subconfluent until the end of the imaging period. After 24 h of incubation, cells were incubated with a complete medium containing 200 ng/ml Hoechst 33342 (Thermo Fisher Scientific) for 1 h. Cells were then transferred to a Nikon Eclipse Ti2 microscope. Images were captured every 12 min with a 20 × 0.8 NA objective for 24 h. Total light exposure time was kept to 200 ms for each time point. Cells were imaged in a humidified 37 °C incubator with 5% $CO_2$. Image processing and cell tracking were carried out with a custom MATLAB script described previously[87]. Live-cell imaging of nuclear size was also performed using the same approach. To observe the effect of GSK126 and JQ1 drugs on nuclear size, 3000 cells were seeded onto Ibidi 96-well polymer coverslip plates. After 24 h, cells were stained with Hoechst 33342 for 45 min, washed the medium three times with PBS, and cells were treated with either 5 μM GSK126 or 1 μM JQ1 and imaged for 24 h.

To study the actomyosin tension dynamics of NUP210-depleted cells, F-tractin-mRuby-P2A-mTurquoise-MLC2 reporter[43], a gift from Dr. Tobias Meyer of Stanford University (Addgene # 85146), was used. 6DT1 cells were transduced with this reporter and 50,000 cells were seeded onto type I collagen (50 μg/ml) coated Ibidi 4-well glass-bottom coverslip. After 24 h, cells were imaged with a Zeiss 880 Airyscan microscope for 1 h (images were taken at every 15 min). Cells were then treated with 1 μM CytoD for 1 h and imaged. After the drug washout, cells were then further imaged (every 15 min) for 15 h in a humidified 37 °C incubator with 5% $CO_2$.

**Cell invasion assay**. The cell invasion assay was performed using BioCoat Matrigel Invasion Chambers with 8.0 μm PET Membrane (Corning) according to the manufacturer's instructions. Briefly, $7.5 \times 10^5$ 4T1 and 6DT1 cells were seeded onto the top well containing DMEM with 0.5% serum. In the bottom well, DMEM with 10% serum was used as a chemoattractant. Cells were incubated in a 37 °C incubator for 24 and 48 h for 6DT1 and 4T1 cells, respectively. After incubation, noninvaded cells were removed from the top well using cotton tips. Cells that had invaded into the Matrigel were then fixed with methanol and stained with 0.05% crystal violet. Matrigel membranes containing invaded cells were then cut and mounted onto glass slides with Vectashield mounting medium (Vector Laboratories). Images of entire membranes were captured as segments and then stitched using an EVOS FL Auto 2 microscope (Invitrogen). Image analysis was performed using the Image-Pro Premier 3D (Media Cybernetics) software. Percent cell invasion was calculated using crystal violet intensity per membrane area.

**CTC analysis**. One hundred thousand 6DT1 cells with or without *Nup210* knockdown were injected into the fourth mammary fat pad of FVB/NJ mice. Ten mice were used in each group and three mice were kept uninjected for use as healthy controls. One month after injection, mice were anesthetized with avertin injection. Through cardiac puncture, 600–1000 μl blood per mouse was collected in 50 μl of 0.5 M EDTA solution. An equal volume of blood was taken for red blood cell lysis using ACK lysis buffer. One hundred microliters of the peripheral blood lymphocyte fraction was subjected to fixation with 2% paraformaldehyde for 15 min at room temperature. Cells were permeabilized with PBS containing 0.1% Triton X-100. Cells were vortexed briefly and kept at room temperature for 30 min. BSA (0.5%) in PBS was added and cells were pelleted by centrifugation. Cells were then resuspended in ice-cold 50% methanol in PBS and incubated for 10 min on ice. One hundred and fifty thousand fixed cells were stained for CD45, a pan-lymphocyte marker, and pan-keratin, a tumor cell marker. Before staining with antibodies, cells were incubated with a FcR blocking reagent (1:10 dilution; Miltenyi) for 10 min at 4 °C. Cells were then stained with APC-conjugated CD45 (1:25 dilution; Miltenyi) and Alexa Fluor 488-conjugated pan-keratin (1:25 dilution, Cell Signaling Technology) antibodies for 10 min at 4 °C. Appropriate isotype control antibodies (Alexa Fluor 488-conjugated mouse IgG1 and APC-conjugated rat IgG2b, k-isotype control) were used along with the primary antibodies. After washing with MACS buffer (PBS, 0.5% BSA, and 2 mM EDTA), cells were

incubated with 1 μg/ml Hoechst 33342 (Thermo Fisher Scientific) for 5 min. Cells were then washed again with MACS buffer and resuspended in 200 μl buffer for analysis using a BD FACSCanto II flow cytometer. A CTC (CD45−/cytokeratin+) gate was created based on the staining pattern of 6DT1 tumor cells in culture and primary tumor cells derived from 6DT1-injected mice. Flow cytometry data were analyzed using the FlowJo V10 software.

**Immunohistochemistry analysis**. Normal human mammary gland tissue section (HuFPT127) and human breast cancer TMA slide (BR10010e) were purchased from US Biomax Inc. (Rockville). Formalin-fixed, paraffin-embedded TMA slides were processed for antigen retrieval for 15 min performed using TAR buffer pH 6 (Dako) in a steamer. Slides were washed followed by endogenous peroxidase (3% hydrogen peroxide in methanol) for 20 min. Protein blocking was performed for 20 min (Dako). NUP210 primary antibody (Atlas) was applied at a dilution of 1:100 in antibody diluent (Dako) for 1 h at room temperature. After incubation, slides were washed in TBS-tween, and Evision+System-HRP-labeled polymer anti-rabbit (Dako) was applied for 30 min at room temperature. Next, slides were washed in TBST, then the signal was developed using 3,3′-diaminobenzidine tetrahydrochloride as the chromogen substrate (Vector Lab). Slides were counter-stained with hematoxylin. A digital image data file was created for the TMA slide by optically scanning with the Aperio AT2 scanner (Leica Biosystems) at 0.25 μm/pixel resolution (×40 objective). HALO (Indica Labs) Artificial Intelligence (AI) neural network software (mini-net, Indica) was utilized to produce a classification algorithm for tumor and stroma area segmentation and quantification. To quantify the NUP210 signal (pixels) in the nuclear envelope of AI-classified tumor cells based upon color and constant image intensity thresholding, a second algorithm was created using HALO Area Quantification to deconvolve colocalization of DAB chromogen and hematoxylin. Once established, the parameters for image processing were held constant across the experiment. Derived labeling ratios were indices produced by dividing immunolabeled areas of nuclear envelope by tumor area (×100).

**Patient datasets analysis**. DMFS analysis on gene expression signature was performed using GOBO tool (http://co.bmc.lu.se/gobo/gobo.pl)[88]. Data on *NUP210* and histone H3.1 (*HIST1H3A, HIST1H3B*) amplification in human breast cancer patients were derived from METABRIC data available on cBioPortal (https://www.cbioportal.org/)[89]. METABRIC *NUP210* mRNA expression data were also downloaded from the cBioPortal database and manually processed for further analysis. DMFS data (*NUP210* mRNA) were obtained from the Km-plotter database (https://kmplot.com/analysis/). The JetSet best probe 213947_s_at was used for *NUP210* expression and patients were separated by upper and lower quartile values. Distant metastasis-free survival data (NUP210 protein) were obtained from Km-plotter database protein module. *NUP210* gene expression data on breast cancer metastatic sites, prostate cancer, and melanoma were downloaded from GEO.

**Statistics and reproducibility**. Experiments have been repeated at least duplicate or triplicates except for studies involving mice (metastasis assay) where the experiment was carried out once for each orthotopic transplantation models to limit the use of animals. Three different immunocompetent orthotopic transplantation models of metastasis and two different mice backgrounds were used to validate the robustness of key results in different biological systems. For in vivo animal studies, *p* values were calculated using the Mann–Whitney *U* test for comparison of two groups in the GraphPad Prism 8 software and the results were reported as mean ± standard deviation. For comparison of more than two groups, analysis of variance (ANOVA) with appropriate multiple comparison tests were performed in GraphPad Prism software. For microscopy image quantification analysis, either Mann–Whitney *U* test (comparison of two groups) or ANOVA with appropriate multiple comparison tests (comparison of multiple groups) were performed. For qRT-PCR result analysis, *p* values were calculated using a multiple two-tailed, unpaired *t* test in GraphPad Prism and results were reported as mean ± standard error of the mean (s.e.m).

**Microscopy image analysis and quantification**. The periphery/total nuclear area intensity ratio of histone H3.1/3.2, H3K27me3, and H3K9me3 was quantified using the Fiji software[81]. Briefly, the total nuclear area was defined by DAPI staining, while the nuclear periphery was defined as the region 25% towards the interior from the nuclear edge (average distance for sg-Ctrl: 0.97 μm and KO-N13: 0.89 μm). Mean fluorescence intensity was quantified in both the periphery and total nuclear area and the periphery/total ratio was plotted. The heterochromatic foci/total nuclear area intensity of histone H3.1/3.2 was quantified using the Fiji software. Heterochromatic foci regions were segmented using DAPI staining. Mean fluorescence intensity was quantified in both heterochromatic foci and total nuclear area and the heterochromatic foci/total ratio was plotted as box plots (bottom and top of the box denote first and third quartile, respectively, whiskers denote ±1.5 interquartile range (IQR), horizontal lines denote median values). *P* values were calculated using a two-tailed Mann–Whitney *U* test.

The position of DNA FISH spots relative to the nuclear centroid and periphery was quantified using the Cell module within the Imaris image analysis suite

(Bitplane). To account for differences in nuclear size, the relative position was calculated by dividing the shortest distance of the FISH spot to the nuclear centroid, by the length of the three-point line encompassing the nuclear centroid, FISH spot, and nuclear periphery. Relative positions were plotted as cumulative distribution frequencies using the RStudio software. $P$ values were calculated using a two-tailed Mann–Whitney $U$ test. The distance of DNA FISH spots to heterochromatic foci was quantified using the Fiji software with the 3D region of interest manager plugin, TANGO[90]. Heterochromatic foci were segmented using DAPI staining and used to create a distance map. DNA FISH spots were segmented and the distance to the nearest heterochromatic foci was calculated and plotted as box plots (bottom and top of the box denote first and the third quartile, respectively, whiskers denote ±1.5 IQR, horizontal lines denote median values). $P$ values were calculated using a two-tailed Mann–Whitney $U$ test.

Focal adhesion counts per cell were quantified using the Fiji software. Cytoplasmic areas were segmented using phalloidin staining. p-FAK Y397 stained focal adhesions were segmented as follows. Adhesions were filtered from image noise using a difference of Gaussian band-pass filter. The filtered image was subsequently segmented using an intensity threshold. Segmented adhesions were counted and quantified on a per-cell basis.

Three-dimensional surface reconstruction of Z-stack images has been performed using Imaris image analysis suite version 9.7.1 (Bitplane).

**Reporting summary**. Further information on research design is available in the Nature Research Reporting Summary linked to this article.

## Data availability

Gene expression (RNA-seq) and ChIP-seq data generated in this study have been deposited in Gene Expression Omnibus (GEO) database under the accession number GSE146591. All the Source data underlying Figures, Supplementary figures, and uncropped western blot images are provided as a Source data file. Supplementary Movie files (1–11) are provided as additional Supplementary Data files. All the other data are available within the article and its Supplementary information. A reporting summary of this article is available as a Supplementary information file. Source data are provided with this paper.

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

## Acknowledgements

The authors thank Sarah Deasy, Ngoc-Han Ha, and Brandi Carofino for critical review of the manuscript, Center for Cancer Research (CCR) Flow Cytometry Facility member Ferenc Livak for FACS analysis, CCR Genomics Core facility member Liz Conner, CCR sequencing facility (Frederick, MD) for RNA-seq/ChIP-seq analysis, and CCR mass spectrometry core facility for LC-MS analysis. The study was supported by the Intra-mural Research Program, National Cancer Institute, National Institutes of Health.

## Author contributions

R.A. conceived the project, performed the majority of the experiments, analyzed the data, and wrote the manuscript. A.S., J.J.Z., S.K., P.W., and S.Z.T. performed chromatin accessibility, ChIA-PET analysis, and wrote the manuscript. S.B. performed DNA FISH analysis. H.L. and M.P.L. performed ChIP-seq/RNA-seq analysis. D.P. and S.D.C. analyzed live-cell imaging data. A.D.T. and M.J.K. performed microscopy image analysis and wrote the manuscript. J.E.D. and R.M.S. analyzed immunohistochemistry data. G.L.H. supervised the chromatin accessibility analysis. Y.R. supervised the ChIA-PET analysis. K.W.H. conceived and supervised the project, analyzed the data, and wrote the manuscript.

## Funding

## Competing interests

The authors declare no competing interests.
