## [Peer Review File · Nature Communications]

Nuclear pore protein NUP210 depletion suppresses metastasis through heterochromatin-mediated disruption of tumor cell mechanical responseEditorial Note: This manuscript has been previously reviewed at another journal that is not operating a transparent peer review scheme. This document only contains reviewer comments and rebuttal letters for versions considered at *Nature Communications*.

REVIEWERS' COMMENTS

Reviewer #1 (Remarks to the Author):

I appreciate the authors' detailed responses to my concerns. While it is desirable to see more robust effects in some of the experiments, I think there are sufficient interesting findings to warrant the publication of this work in *Nature Communications*.

Reviewer #4 (Remarks to the Author):

I am a late joiner to the review team for this paper, and have been asked specifically to give my consideration to the response to the earlier comments of reviewer 2. It is my general opinion that this is an interesting paper that is accompanied by an extensive quantity of data, and that it appears to have been subjected to a thorough process of review. Reviewer 2 was right to flag that substantial and significant changes in lamin-A/C (Fig. 7f) could have important consequences on cell behaviour. The measurements that were originally presented were based on quantitative immunofluorescence. These could reflect true protein concentrations, or could perhaps be due to protein conformational changes / epitope effects (Ihalainen et al. *Nat Mater.* 2015), or an artefact in microscopy-based quantification caused by concurrent changes in nuclear morphology. A reasonable follow-up, therefore, was to check lamin-A/C levels by Western blot – this data is now included in the supporting information, showing no significant change.

In their response to review, the authors suggest that further investigations into the role of lamin-A/C have exciting potential. I would agree with this, but think that the extent of investigation currently presented is a justifiable end-point for this manuscript. The response letter speculates on how lamin-A/C might be connected to their other observations, but to fully explore these possibilities would require extensive additional study (that I would consider beyond the scope of this work). Lamins are relatively highly expressed and function in multimeric, structured assemblies. As the authors note in response to reviewer 1, relatively small % changes in protein concentration could therefore be functionally important e.g. representing a sizeable investment of cellular resources, or significantly affecting the function of the lamina. The same points could be made for the effects of post-translational modification. Lamins are also noted for interacting with a range of binding partners within the cell, such that a change in lamin level is likely to be accompanied by wide-scale remodelling of the proteome (Note: this remodelling could affect actin, used for normalisation in supporting figure 7a). This issue could be addressed experimentally through -omic approaches, but it would remain challenging to establish causation (for example, as a singular contributor to mechanical properties). This complexity reflects in the fact that, despite the animal study cited by reviewer 2, there is no characteristic observation in lamin levels that applies across all cancers (Irianto et al. *Cell Mol Bioeng.* 2016).